# Aerosol arriving on the Caribbean island of Barbados: Physical properties and origin

Heike Wex[1], Katrin Dieckmann[1,*], Greg C. Roberts[2,3], Thomas Conrath[1], Miguel A. Izaguirre[4], Susan Hartmann[1], Paul Herenz[1], Michael Schäfer[1,**], Florian Ditas[1,***], Tina Schmeissner[1], Silvia Henning[1], Birgit Wehner[1], Holger Siebert[1], and Frank Stratmann[1]

[1]Leibniz Institute for Tropospheric Research, Experimental Aerosol and Cloud Microphysics, Leipzig, Germany.
[2]Centre National de Recherche Scientifique, Meteo France, Toulouse, France.
[3]Scripps Institution of Oceanography, Center for Atmospheric Sciences, La Jolla, United States.
[4]Meteorology and Physical Oceanography, University of Miami RSMAS, Miami, USA.
[*]now at Eurofins GfA GmbH, Münster, Germany
[**]now at Leipzig Institute for Meteorology, University of Leipzig, Leipzig, Germany
[***]now at Max Planck Institute for Chemistry, Mainz, Germany

*Correspondence to:* H. Wex (wex@tropos.de)

**Abstract.** The marine aerosol arriving at Barbados (Ragged Point) was characterized during two three-week long measurement periods in November 2010 and April 2011, in the context of the measurement campaign CARRIBA (Cloud, Aerosol, Radiation and tuRbulence in the trade wInd regime over BArbados). By comparison between ground based and airborne measurements it was shown that the former are representative for the marine boundary layer at least up to cloud base. In general, total particle number concentrations ($N_{total}$) ranged from as low as 100 cm$^{-3}$ up to 800 cm$^{-3}$, while number concentrations for cloud condensation nuclei ($N_{CCN}$) at a supersaturation of 0.26% ranged from some ten to 600 cm$^{-3}$. $N_{total}$ and $N_{CCN}$ depended on the air mass origin. Three distinct types of air masses were found. One type showed elevated values for both, $N_{total}$ and $N_{CCN}$ and could be attributed to long range transport from Africa, by which biomass burning particles from the Sahel region and / or mineral dust particles from the Sahara were advected. The second and third type both had values for $N_{CCN}$ below 200 cm$^{-3}$, and a clear minimum in the particle number size distribution (NSD) around 70 to 80 nm (Hoppel minimum). While for one of these two types the accumulation mode was dominating (albeit less so than for air masses advected from Africa), the Aitken mode dominated the other and contributed more than 50% of all particles. These Aitken mode particles likely were formed by new particle formation no more than three days prior to the measurements. Hygroscopicity of particles in the CCN size range was determined from CCN measurements to be $\kappa = 0.66$ on average, which suggests that these particles contain mainly sulfate and do not show a strong influence from organic material, which might generally be the case for the months during which measurements were made. The average $\kappa$ could be used to derive $N_{CCN}$ from measured number size distributions, showing that this is a valid approach to obtain $N_{CCN}$. Although the total particulate mass sampled on filters was found to be dominated by Na$^+$ and Cl$^-$, this was found to be contributed by a small number of large particles ($> 500$ nm, mostly even in the super-micron size range). Based on a three-modal fit, a sea spray mode observed in the NSDs was found to contribute 90% to the total particulate mass, but only 4 to 10% to $N_{total}$ and up to 15% to $N_{CCN}$. This is in accordance with finding no correlation between $N_{total}$ and wind speed.

# 1 Introduction

Atmospheric aerosol particles can act as cloud condensation nuclei (CCN) on which cloud droplets form, and hence there is a connection between atmospheric aerosol particles and clouds. Already *Twomey* (1974) and *Albrecht* (1989) examined respective interactions. *Twomey* (1974) described that an increase in CCN leads to the formation of more but smaller cloud droplets, assuming a constant liquid water content in the examined clouds. This may delay the formation of precipitation, which then could increase cloud lifetime, as proposed by *Albrecht* (1989). Today, many more aspects of aerosol-cloud interactions have been described, but we still lack understanding of the overall roles of aerosol particles, clouds and their interactions in the climate system. This fact manifests in the following statement in Chapter 7 ("Clouds and Aerosols") of the latest report by the International Panel on Climate Change, *Boucher et al.* (2013): "Clouds and aerosols continue to contribute the largest uncertainty to estimates and interpretations of the Earths changing energy budget."

The first step in the formation of warm and mixed-phase clouds is the activation of aerosol particles to cloud droplets. At temperatures below $0°C$, freezing complicates the processes ongoing in clouds. Above $0°C$ only warm clouds exist, in which, after the activation of aerosol particles, a number of processes influence the further development of the cloud and its droplets, among them coagulation, condensation, and entrainment and mixing at the cloud edges. An influence of atmospheric aerosol particles and particularly of CCN on clouds is discussed in context of the droplet number concentration, the droplet size, the amount of drizzle forming from a cumulus cloud and the cloud coverage (e.g., *Heymsfield and McFarquhar*, 2001; *Conant et al.*, 2004; *Hudson and Mishra*, 2007; *Kaufman et al.*, 2005). Particularly the role of aerosol particles in the formation of precipitation remains controversial (*Stevens and Feingold*, 2009), as meteorological conditions might be of large importance for these processes, too. To shed further light on these issues, it is important to make atmospheric measurements under meteorological conditions which are as stable as possible, but with changing loads of atmospheric aerosol particles. These requirements are met in the trade wind region upwind of Barbados. Additionally, except for ship and aircraft emissions across the Atlantic, there are no anthropogenic particle sources upwind of Barbados for several thousands of kilometers.

The marine aerosol in general and in the Caribbean in particular has been studied before. However, much that is known, particularly for the Caribbean, is based on remote sensing, satellite data, modeling results or scarce data points taken onboard of aircraft, e.g., during RICO (Rain In Cumulus over the Ocean, *Rauber et al.*, 2007), or BACEX (Barbados Aerosol Cloud EXperiment in 2010, *Jung et al.*, 2016). Concerning long term studies, often filter samples were examined (e.g., *Novakov et al.*, 1997; *Prospero and Lamb*, 2003; *Peter et al.*, 2008), but size-segregated in-situ aerosol data for marine aerosol particles in the Caribbean are still scarce.

Aerosol particles from a marine source, often called sea spray or sea salt particles, are generally produced in dependence on surface wind speeds over oceans (e.g., *O'Dowd and de Leeuw*, 2007), where the notation sea spray refers to the possibility that also organic material might be included in these particles. They can be expected to be produced also directly in the vicinity of Barbados. These particles have a high hygroscopicity and as they are produced in moist surroundings at the lowest levels of the atmosphere, they have comparably short life times (*Textor et al.*, 2006). Hygroscopic growth measurements onboard of ships identified sea salt particles in the Pacific and Southern Oceans (*Berg et al.*, 1998), in the Atlantic and Indian Ocean

(*Massling et al.*, 2003) and in the Western Pacific off the coast of Asia (*Massling et al.*, 2007). The occurrence of these particles was related to wind speeds above 10 m s$^{-1}$ (*Berg et al.*, 1998; *Massling et al.*, 2003) and larger particle sizes (found at 250 and 350 nm in rare clean marine cases, but mostly only at 1000 nm, *Massling et al.*, 2007). *Bates et al.* (1998), participating in the same Southern Ocean ship cruise mentioned before, describe the presence of sea salt particles in a coarse particle

mode, saying that they might dominate the mass size distribution but only contribute little to number concentrations. More recently, *Modini et al.* (2015) examined marine particle number size distributions also obtained onboard a ship off of the coast of California, US. They applied three-modal fits and attributed the third mode, visible as a shoulder at particles sizes > 500 nm, to sea spray aerosol. During a phase of high wind speeds of 16 m s$^{-1}$, these particles from sea spray contributed on average 16 to 28% of all CCN at 0.3% supersaturation, while at lower wind speeds of 12 m s$^{-1}$, this fraction decreased to 5 to 10%.

*Modini et al.* (2015) concluded that under clean marine conditions, when the ocean surface is relatively calm, particles from sea spray contribute very little to marine CCN number concentrations. Comparable results were also obtained based on modeling: *Glantz et al.* (2004) retrieved steep vertical gradients in the sea salt mass concentrations over the Atlantic, and suggest that sea salt particles do not significantly contribute to CCN. This is in agreement with a modeling study of *Dunne et al.* (2014), where it was found that over oceans, wind speed changes and hence aerosol particles from sea spray had an negligible effect

on marine CCN, and that overall, wet removal of CCN by nucleation scavenging played a dominant role in regulating marine CCN concentrations.

It has been assumed in the past, that a few large particles produced from sea spray, so called giant nuclei (GN), could play an important role in the warm rain process, i.e., in the rapid formation of precipitation in trade wind cumuli (see e.g., reviews by *Beard and Ochs*, 1993; *Blyth et al.*, 2013). This was already examined by *Woodcock* (1953) and is still a topic of

active research. *Rudich et al.* (2002) found that at the continental region around the Aral Sea large salt-containing dust particles increase cloud drops to sizes that promote precipitation. *Teller and Levin* (2006) and *Hudson and Mishra* (2007) found through modeling and based on aircraft measurements done in the Caribbean, respectively, that GN have little influence on precipitation when CCN concentrations are low. *Blyth et al.* (2013) also claim that GN are unimportant for the warm rain process in shallow, maritime clouds and suggest in addition that this process can be modeled relatively simply based on condensational growth

of droplets formed on sub-cloud aerosol particles. Overall, the role and importance of sea spray particles are still discussed controversially.

Similarly, the contribution of organic compounds to marine aerosol particles is under discussion. *Novakov et al.* (1997) collected aerosol particles < 600 nm in the marine trade wind region on Puerto Rico and found that they contained a fraction of soluble organic material which exceeded that of sulfates. Generally, the ocean has been assumed to be a source for organic

particulate matter, a topic on which a review can be found in *Gantt and Meskhidze* (2013). It has therefore been speculated that the organic fraction of the marine aerosol has an influence on number concentrations of particles and CCN and on their hygroscopicity.

On the other hand, in a modeling study by *Korhonen et al.* (2008) a seasonal cycle observed for CCN concentrations at Cape Grim, Tasmania, could be connected to a very similar seasonal cycle for atmospheric dimethyl sulfide (DMS). *Korhonen et al.*

(2008) claim that the observed CCN are formed by nucleation of DMS-derived $H_2SO_4$ in the free troposphere followed by

subsequent condensational growth and coagulation, which takes several days to over a week. When CCN are finally entrained into the marine boundary layer (MBL), they may be hundreds or even thousands of kilometers away from the site of the original DMS emissions. Similarly based on a global aerosol microphysics model, *Merikanto et al.* (2009) found that in the marine boundary layer, 55% of all particles acting as CCN at a supersaturation of 0.2% originate from new particle formation, with 45% being entrained from the free troposphere and the remaining 10% nucleating in the boundary layer directly. Ammonium sulfate was reported as the main component of aerosol particles with diameters below 400 nm for marine aerosol sampled in the Caribbean by *Peter et al.* (2008), while larger particles were found to mainly consist of sea salt. These results are consistent with those of *Modini et al.* (2015), who state that the largest contribution to CCN originates from particles in the accumulation mode centered at 100 to 200 nm, of which non-sea-salt sulfate was the controlling chemical component. *Allan et al.* (2008) examined aerosol particles on Puerto Rico, and air masses arriving from east-northeast were found to be mostly free of anthropogenic influences. The sub-micron fraction of aerosol particles in these air masses was mainly composed of non-sea-salt sulfate, and organic compounds did not play an important, if any, role.

Early on in atmospheric aerosol research, *Prospero et al.* (1970) discovered that Saharan dust can be transported to Barbados and the Caribbean in general, a discovery which showed that atmospheric aerosol particles can be transported over long distances. The dust reaches the Caribbean in lofted layers called saharan air layer (SAL, *Smirnov et al.*, 2000), but it can also be measured at the surface throughout the year, albeit in an annual cycle with a maximum in summer (*Prospero et al.*, 1981; *Prospero and Lamb*, 2003). Particles originating from biomass burning in Africa are also frequently transported into the Caribbean and particularly in winter this biomass burning aerosol can occur embedded in the dust layer (e.g., *Haywood et al.*, 2004; *Kaufman et al.*, 2005; *Ansmann et al.*, 2011). Using a global aerosol model, *Dunne et al.* (2014) investigated the aerosol processes determining marine CCN concentrations, and found that even in the remote Southern Oceans, long-range transport of continental aerosol influences day-to-day fluctuations and the highest concentrations of CCN.

Overall, there are sources for more and less hygroscopic particles which might contribute to the aerosol arriving in the Caribbean. *Pringle et al.* (2010) used an atmospheric chemistry model to derive global values for CCN hygroscopicity, represented as $\kappa$ (*Petters and Kreidenweis*, 2007). Annual mean values at the surface of Barbados were derived to be roughly 0.55, with an annual cycle ranging from 0.3 in September to 0.7 in December. For April and November, the months during which the here presented measurements were made, values of 0.65 and 0.5 were reported, respectively. Herein, these modeled values will be compared to those obtained from in-situ measurements during our campaigns in the framework of CARRIBA (Cloud, Aerosol, Radiation and tuRbulence in the trade wInd regime over BArbados) and also during SALTRACE (Saharan Aerosol Long-range Transport and Aerosol-Cloud-Interaction Experiment), another campaign which recently took place on Barbados (*Kristensen et al.*, 2016), in which comparably low values of 0.2 to 0.5 were found for the summer months of June and July.

To tackle open issues around warm clouds, the CARRIBA-project was initiated by the Leibniz Institute for Tropospheric Research (TROPOS). The easternmost island in the Caribbean, Barbados, was used as a base to examine the interplay between undisturbed marine aerosol and warm clouds. Ground based and air-borne measurements were performed, where for the latter the measurement platform ACTOS was deployed (Airborne Cloud Turbulence Observation System, *Siebert et al.*, 2006). An

overview over the measurement activities, which took place during the two periods of CARRIBA in November 2010 and April 2011, is given in *Siebert et al.* (2013).

In the present study, we focus on a thorough in-situ characterization of the aerosol with respect to number concentrations of particles, particularly CCN and particle number size distributions, both on ground and in the MBL below cloud base, i.e., the sub cloud layer (SCL). We will show that the measurement station at Ragged Point, Barbados, allows for the characterization of atmospheric aerosol representative of the SCL. We will discuss contributions of different materials (sea salt, organics, sulfates, mineral dust and biomass-burning) to both CCN number concentrations and hygroscopicity. And we will show that the marine aerosol transported into the Caribbean can be roughly categorized into three classes, where one can be considered clean marine (with comparably low CCN and total particle number concentrations), one as influenced by new particle formation, and the last one as influenced by continental emissions, mostly from Africa, advected across the Atlantic. Aerosols arriving on Barbados often have been considered clean and/or typical marine, due to the incoming air masses being free of anthropogenic influences for thousands of kilometers prior to arrival. But we will show that even this aerosol is not free of continental, and hence possibly anthropogenic, influences, as already suggested by *Hamilton et al.* (2014) who claim that pristine atmospheric aerosols today are mainly only found on the Southern Hemisphere and only very rarely in the Northern Hemisphere.

## 2   Measurements

Measurements in the framework of CARRIBA took place during November 2010 and April 2011 on Barbados. Barbados is the easternmost island of the Caribbean, with prevailing easterly winds. The respective air masses arriving at the island have been free from continental and also from anthropogenic influences (except for ship and aircraft emissions) for some thousand kilometers and hence for some days, and the aerosol arriving at Barbados can be assumed to represent typical marine aerosol. In the following, details will be given about the ground based measurements, followed by information about the air-borne measurements done with ACTOS.

Ground based measurements were carried out at Ragged Point on Barbados, on one of the easternmost tips of the island. Ragged Point is the location where Joe Prospero and his group intensively investigated Saharan dust transport into the Caribbean (e.g., *Prospero and Nees*, 1986; *Prospero and Lamb*, 2003). We used the measurement container, together with its 17 m high mast towering over the 30 m high ragged cliffs. On top of the mast, daily filter samples were taken on Whatman-41 filters, sampling roughly 670 l min$^{-1}$. Sampling and subsequent analysis was as described in e.g., *Savoie et al.* (1989) and *Savoie et al.* (2002): After sampling, the filters were folded, stored in food grade plastic bags under a clean hood, and once a week the collected samples were sent to the University of Miami where they were stored at -4° before they were analyzed. The analysis was done with respect to their mass content of major water-soluble ions (Cl$^-$, Na$^+$, SO$_4^{2-}$, NO$_3^-$, K$^+$, Ca$^{2+}$) and ash. The latter was what remained of the insoluble residual after placing the filter in a furnace at 550° for 14 hours, and it can be attributed to particles from desert dust or biomass burning, distinguishing between the two based on blackness.

An aerosol inlet had been installed on top of the mast, i.e., on top of a 17 m long tube. The tube had a diameter of 8 inches ($\approx 0.2$ m), and a (laminar) total flow of 350 l min$^{-1}$ was drawn through it. This comparably high flow rate was used to shorten

the residence time in the tube to $\approx 90$ s. The aerosol used for the measurements was sampled from the center of this tube at its lower end and was led into the measurement container. A cyclone was installed where the sampling tube entered the container. This was done to enable the inversion of the size distributions, which could only be measured up to a dry diameter of 500 nm. The cyclone was operated with a flow rate of $4\,l\,min^{-1}$, where a by-pass flow of $\approx 2.7\,l\,min^{-1}$ was purged directly following

the cyclone. The 50% cut off of the cyclone was at a dry diameter of $\approx 275$ nm. Particle losses in the cyclone for diameters up to $\approx 200$ nm were negligible, and all particles $> 500$nm were removed. The efficiency of the cyclone, the related loss of particles and related necessary corrections are discussed in the Appendix (App. A3).

For the aerosol being fed into the instruments, different drying systems had been installed in-line. First the aerosol was led through two Nafion tubes. The pressure outside the tubes was kept at 300 mbar to remove some of the water vapor. The

10 two dryers were followed by a diffusion dryer filled with silica gel. The relative humidity (RH) after the passage through the diffusion dryer was measured continuously by drawing $0.2\,l\,min^{-1}$ of the aerosol past a hygrometer (B+B Thermo-Technik, Hytelog RS232). Once the RH was above 25%, the diffusion dryer was replaced by a new one filled with dry silica gel. The replacement had to be done roughly every two days.

The dried aerosol flow was fed into the instruments. An overview of the instrumentation used at Ragged Point and on

ACTOS can be seen in Table 1. A Kr-85 neutralizer was installed in front of a DMA (Differential Mobiliy Analyzer, Type Vienna Hauke medium, aerosol to sheath air flow ratio 1:10), and the DMA was used to select particle sizes. The aerosol flow containing the size selected particles was then fed into a CPC (Condensation Particle Counter, TSI 3010) and a CCNC (Cloud Condensation Nucleus Counter, Droplet Measurement Technologies (DMT), *Roberts and Nenes*, 2005). CPC and CCNC both were operated with an aerosol flow rate of $0.5\,l\,min^{-1}$. While this is the standard operation flow rate for the CCNC, this is not

so for the CPC, and a calibration of the detection efficiency of the CPC at this flow rate had been done at the home laboratory and was included in the data evaluation. At the DMA, 30 logarithmically equidistant distributed diameters from 25 nm to 500 nm were adjusted, each for 1 minute. Using the combination of DMA and CPC, i.e., a DMPS (Differential Mobility Particle Sizer), particle number size distributions (NSDs) were measured with a time resolution of 30 minutes. Measured size distributions were inverted based on the charge distribution as described by *Wiedensohler* (1988). The supersaturations

set at the CCNC were changed after each distribution and were subsequently set to 0.07%, 0.1%, 0.2%, 0.4% and 0.7% (in April 2011, also 0.3% was used, additionally). A mini-CCNC was also used to measure total (also known as polydisperse) CCN number concentrations ($N_{CCN}$). It was operated at a constant measurement flow rate of $0.1\,l\,min^{-1}$ while continuously scanning the temperature difference over the activation column, set for measuring in a range of supersaturations from 0.08% up to 0.65% in November 2010 and up to 0.8% in April 2011. The supersaturations of both, the DMT- and the mini-CCNC,

were calibrated at the beginning and the end of the measurements campaigns at Ragged Point.

Particle losses in the whole inlet system were either calculated, based on tubing length and diameters or, in the case of the drying system and the cyclone, were determined in the home laboratory (for details see App. A1 and A3). The measurements were corrected respectively. Additionally, all particle and CCN number concentrations reported in this study are given for standard pressure and temperature (STP). Altogether, for the ground based measurements, roughly three weeks of data were

35 obtained for each of the two months.

Airborne measurements were done on ACTOS. ACTOS is known for enabling measurements with a high spatial resolution. It is hung underneath a helicopter (fixed with a 140 m long rope), which typically travels with a true airspeed of 20 m s$^{-1}$. During CARRIBA, the helicopter carrying ACTOS was operated from a helipad located at the Grantley Adams International Airport on Barbados. The main operational area for ACTOS flights was an area of roughly 100 km$^2$ upwind of the island, above

the open Atlantic. 17 and 16 research flights, each lasting roughly two hours, were done in November 2010 and April 2011, respectively. Flights mostly started between 1 pm and 3 pm UTC (9 am and 11 am local time), while a quarter of all flights were afternoon flights, going out as late as 7:30 pm UTC (3:30 pm local time). Including the time span of the flights, measurements on ACTOS were done during times covering 8 hours altogether, all during day light. Due to comparably stable sea surface temperatures which show no strong diurnal variations, the marine boundary layer shows no strong diurnal variation either, in

contrast to what is observed for the continental boundary layer. Therefore, measurements taken for the here presented study can be seen as representative for the marine aerosol in the Caribbean, independent of the time of day during which they took place.

A detailed description of the instrumentation flown on ACTOS during CARRIBA can be found in *Siebert et al.* (2013). In the present study, ACTOS data from the following instruments was used (for an overview see Table 1): A CPC (TSI 3762)

which was operated with a temperature difference between saturator and condenser of 25 K measured total particle number concentrations ($N_{total}$) for particles larger than 6 nm with a measurement frequency of approximately 1 Hz. NSDs were measured by a SMPS (Scanning Mobility Particle Sizer, *Wehner et al.*, 2010) providing a full spectrum between 6 and 230 nm at a time resolution of 2 min. For larger aerosol particles an OPC (Optical Particle Counter 1.129 (SKY-OPC), Grimm Aerosol Technik GmbH) measured NSDs in the range of 250 to 2500 nm at a measurement frequency of 1 Hz. This OPC was calibrated

with spherical latex particles with a refractive index of m = 1.586 + i0.0, and the particle number size distributions were derived using the refractive index representative for ammonium sulfate (m = 1.53 + i0.0). Also, a mini-CCNC similar to the one operated at Ragged Point was used for measuring $N_{CCN}$. All data from the mini-CCNC on ACTOS reported in the present study were taken at a supersaturation of 0.26%. This mini-CCNC was calibrated prior to, during and after the measurement campaigns, similar to the procedure applied to the CCNCs used at Ragged Point. All aerosol instrumentation was situated in the

body of ACTOS and was fed with aerosol through a sampling line. A silica gel diffusion dryer dried the aerosol to < 40% RH before it was distributed to the instruments.

For the measurement of cloud droplet size distributions, a Phase Doppler Interferometer (PDI) for single droplet measurements in the size range of 1 to 180 $\mu$m was used (*Chuang et al.*, 2008, Atrium Technologies,). The PDI was mounted on the outside, in front of ACTOS, to enable the detection of cloud droplets in-situ.

We end this overview of the measurements with mentioning that also all data measured on ACTOS were loss corrected and are reported for STP conditions throughout this work.

# 3 Results

## 3.1 Particle number size distributions

Three different types of NSDs were found time and time again. Figure 1 exemplarily shows these three different types as measured on ground at Ragged Point (solid lines) and simultaneously on ACTOS (dotted lines) in the SCL. Data were taken on Day Of Year (DOY) 319 (black lines), 328 (green lines) and 329 (red lines) between roughly 1 pm and 3 pm UTC. For each flight all NSDs measured on ACTOS in the SCL between 100 and 400 m were averaged, where it shall be mentioned that cloud base was often observed at 500 m (*Siebert et al.*, 2013), so data included here was selected conservatively. Similarly, all NSDs measured at Ragged Point for the duration of the respective ACTOS flight were averaged as well. Losses were accounted for as described in App. A1, while for the here shown NSDs from Ragged Point corrections for losses in the cyclone as described in App. A3 were not performed for illustrative reasons.

The NSDs shown in Figure 1 are representative for the whole duration of the CARRIBA campaigns. Three distinctly different particle modes were detected in almost all cases during both months in which we measured. The overall shape of the NSDs measured on ground agreed with that of the NSDs measured in the SCL up to approximately 200 to 300 nm. Measured number concentrations for diameters below roughly 70 nm were generally slightly larger in the ground based data-set, compared to data taken on ACTOS, possibly due to an overestimation of losses in the diffusion dryers (see App. A1). For diameters above approximately 200 to 300 nm, number concentrations measured at Ragged Point were below those measured on ACTOS. This is exceedingly so for increasing diameters up to 500 nm, which is the upper diameter for which measurements at Ragged Point were made. Here, the effect of the cyclone that was installed in the inlet system at Ragged Point becomes visible. Generally, there is good agreement between NSDs measured on ground and in the SCL in the size range where the cyclone did not yet effect the measurements. This is a first hint towards the fact that the SCL was well mixed with respect to both heights and also regional extent, and that the inlet mast at Ragged Point indeed sampled air from the SCL without much influence from the surf at the cliffs of Ragged Point, at least for the size range investigated.

In the following, the three distinct types of NSDs shown in Figure 1 will be discussed. NSDs resembling those shown in black (panel A) show three modes, an Aitken-, an accumulation- and a sea spray mode, which can clearly be distinguished. These NSDs will be attributed to the "marine-type" in this work. The minimum between the Aitken- and accumulation mode of the NSDs (Hoppel minimum, see *Hoppel et al.*, 1986) at roughly 70 nm to 80 nm indicates the sizes above which particles had previously been activated to cloud droplets during the history of the air mass at least once. While passing through a cloud, soluble material is added to the activated particles by wet phase chemistry, increasing particulate mass and hence also the size of these particles.

During some times NSDs were observed which had a very pronounced Aitken-mode with a maximum at about 30 nm, as seen in Figure 1 (panel B, green curves). For these NSDs, the Hoppel minimum was clearly visible as well. The respective NSDs will be attributed to the "Aitken-type" herein. The observed small particles can be assumed to originate from new particle formation. They could have been formed in the free troposphere long distances away, as discussed in *Korhonen et al.* (2008). Growth rates observed in the tropical and subtropical MBL were reported to be roughly 1 to 6 nm per hour in a review by *Kulmala et al.*

(2004). Based on measurements made during the CARRIBA campaigns, *Wehner et al.* (2015) reported additional events of new particle formation with much higher growth rates up to nanometers per minute for particles with diameters of roughly 10 nm. These events occurred just outside of clouds, in isolated air parcels with an extent of several ten meters. Particles of both origins, formed in the free troposphere or just around the trade wind cumuli, may have added to the observed small particles.

5 Overall, the age of particles in the Aitken-mode can be estimated to be on the order of a few hours up to a maximum of three days. While particles in this size range were always present, they made up more than 50% of all particles for Aitken-type NSDs.

Finally, displayed in red in Figure 1 (panel C), during some times we observed an increase in number concentrations caused by particles in the accumulation-mode size range. During these events, $N_{CCN}$ increased above 200 cm$^{-3}$ (as discussed in 10 more detail in Sec. 3.2) and the accumulation-mode could become so large that the Hoppel minimum was not clearly visible any more. These NSDs will be attributed to the "accumulation-type". Later on (see Sec. 3.3), the three different types of NSDs will be correlated to different air masses observed in this study.

It should be mentioned that already *Bates et al.* (1998) report that NSDs observed during ship measurements in the Southern Ocean fell into three distinct categories, similar to those observed in this study. More recently, based on air borne aerosol in-15 situ measurements done in the vicinity of Barbados, *Jung et al.* (2016) also mention that NSDs were consistent with air mass origin, where, however, NSD measurements were only done down to particle sizes of 100 nm, and a thorough study on these NSDs has yet to be introduced.

The third mode that was visible in the NSDs measured on ACTOS can be attributed to particles originating from sea spray, in accordance with former research (e.g., *Bates et al.*, 1998; *Modini et al.*, 2015). Fitting three different modes to the NSDs 20 shown in Figure 1 reveals that particles in the third mode contribute $\approx$ 90% of the total particulate mass, but only 4 to 10% of $N_{total}$ and up to 15% of $N_{CCN}$ (i.e., when omitting the Aitken mode), comparable to what was found in older studies (*Bates et al.*, 1998; *Modini et al.*, 2015).

## 3.2 Particle hygroscopicity and number concentrations of CCN

In the following, the particle hygroscopicity is determined, expressed as $\kappa$ (*Petters and Kreidenweis*, 2007), where $\kappa$ can be 25 interpreted as representing the average particle chemistry. Data from measured NSDs and CCN number size distributions were used to obtain activated fractions as a function of particle size. At each diameter at which number concentrations of particles and CCN were measured during a size scan, the activated fraction was determined. The activated fraction curves were corrected for multiply charged particles (see *Deng et al.*, 2011), and an error function fit was done to obtain the critical diameters for activation, i.e., the diameter at which 50% of all particles were activated. These critical diameters, together with 30 the supersaturations at which they were determined, were then used to derive values for $\kappa$, assuming the surface tension to be that of water. Therefore, one measured NDS results in one pair of values for critical diameter and $\kappa$. On average, the critical diameters were 179 nm, 148 nm, 81 nm, 61 nm, and 55 nm (with a standard deviation of $\approx$ 8%) at the set supersaturations of 0.07%, 0.1%, 0.2%, 0.3% and 0.4%.

The retrieved $\kappa$-values scattered much. No clear trends in $\kappa$ (or in critical diameters) could be seen when data were separated according to different air masses, i.e., to the different types of NSD (not shown here; for a general discussion of the different air masses see Sec. 3.3). Also no trends in $\kappa$ were observed when values obtained at different supersaturations were examined separately (as shown in Figure 2, where error bars give the standard deviation for averaging all respective $\kappa$ values). Only for a time roughly from DOY 103 to 107 during April 2011, i.e., during a very distinct dust period (see Sec. 3.3), $\kappa$ was noticeably lower (0.56 on average) than during all other times (0.68 on average). However, when considering the scatter observed in $\kappa$ (see the error bars in Figure 2), these lower values still agree with the others within uncertainty. It should be mentioned here, that uncertainties in the supersaturation adjusted in the CCNC (standard deviation of 3.3% (relative) for supersaturation $\geq 0.1\%$ and $0.07\pm0.0033\%$) could at maximum contribute only 1/3 to this observed scatter.

Overall, $\kappa$ averaged to 0.66. These values are comparable to results from model calculations given in *Pringle et al.* (2010), where $\kappa$ at the surface for the region around Barbados was reported to be 0.65 and 0.5 for April and November, respectively. *Pringle et al.* (2010) furthermore reported a yearly cycle, with lower $\kappa$ of $\approx 0.4$ in June and July. And indeed, *Kristensen et al.* (2016), found lower values for $\kappa$ of 0.2 to 0.5 for June and July 2013 when measuring at Ragged Point on Barbados. Figure 3 shows the yearly cycle of $\kappa$ taken from *Pringle et al.* (2010) together with values derived in *Kristensen et al.* (2016) and in this study. Within measurement uncertainty, there is good agreement, and the yearly cycle derived in *Pringle et al.* (2010) is corroborated by the measurements.

The derived $\kappa$-values suggests that the majority of the particles in the size range between roughly 50 nm up to 200 nm, i.e., those from which $\kappa$ was determined, did not originate from sea spray, as $\kappa \approx 1$ would be expected for sea salt particles (see e.g., *Wex et al.*, 2010b). Instead, $\kappa \approx 0.66$ hints more towards the presence of sulfates (see examples given in *Petters and Kreidenweis*, 2007), which generally are formed during new particle formation and wet phase chemistry. Organic compounds generally have low values of $\kappa$ on the order of 0.1 (*Petters and Kreidenweis*, 2007), and hence were likely not present in the particulate matter during November 2010 and April 2011. But the lower $\kappa$ reported by *Kristensen et al.* (2016) indicate that a larger fraction of the particulate matter consisted of organic compounds in the summer month.

The above mentioned diameter range of 70 to 80 nm for which the Hoppel minimum was observed can be used, together with the average $\kappa$ of 0.66, to obtain a rough estimate of maximum supersaturations present in trade wind clouds along the path of the sampled air masses. Resulting values are roughly 0.2 to 0.25%. This is slightly lower but still close to an earlier estimate given in *Clarke et al.* (1996) of 0.35% and can be interpreted as typical value for trade wind cumuli.

At the supersaturation where most of the mini-CCNC data on ACTOS were taken, at 0.26%, the derived average $\kappa$ of 0.66 corresponds to a critical diameter of 68 nm. It is known that particle size matters more than the chemical composition of the particles (the latter being represented by $\kappa$, here) in determining the critical supersaturation that is needed for a particle to activate (*Dusek et al.*, 2006). It has been shown before for remote continental data (*Juranyi et al.*, 2010) that $N_{CCN}$ can be derived based on time resolved measurements of NSDs and an average $\kappa$ for the examined region. To test if this was also feasible for the marine aerosol examined in the present study, in a next step, $N_{CCN}$ was calculated by integrating the NSDs beginning from the critical diameter of 68 nm onward, and was compared to $N_{CCN}$ measured by the mini-CCNC at a supersaturation of 0.26% ($\pm$ 0.03%).

In this comparison, data obtained from measurements at Ragged Point and on ACTOS were included. Therefore, the ground based data had to be corrected with respect to losses in the inlet system (see App. A1 and A2), and to losses of large particles ($> 200$ nm, see App. A3) which were missing due to the use of a cyclone in the inlet system. Resulting $N_{CCN}$ can be seen in Figure 4, showing values which were derived from measured NSDs from Ragged Point ($N_{CCN}^{RP,NSD}$) and from ACTOS ($N_{CCN}^{A,NSD}$) and which were measured directly at Ragged Point ($N_{CCN}^{RP,CCNC}$) and on ACTOS ($N_{CCN}^{A,CCNC}$). (See Table 1 for a explanatory summary of the symbols used here.)

There is a good correlation of $N_{CCN}$ measured directly to those obtained from the NSDs. This can also be seen in Figure 5 which shows scatterplots comparing the data-sets shown in Figure 4. The most complete data-set exists for $N_{CCN}^{RP,NSD}$, therefore this data-set was used as the base for the comparisons shown in the scatter plots. The two data-sets obtained from ground based measurements are compared in panel A of Figure 4. The agreement in November 2010 is excellent, resulting in a slope of a linear fit of 0.99 and a $R^2$ of 0.95. For the respective data taken in April 2011 a somewhat larger scatter can be seen, but overall, combining both data-sets, a linear fit with a slope of 0.97 and a $R^2$ of 0.91 is obtained. When comparing $N_{CCN}^{RP,NSD}$ to data measured on ACTOS, the ground based data were averaged for the duration of the respective ACTOS flight. The respective data are shown in panel B of Figure 4. The comparison of $N_{CCN}^{RP,NSD}$ to data from the mini-CCNC on ACTOS resulted in a linear fit with a slope of 1.00 and a $R^2$ of 0.86, i.e., also for this comparison a very good agreement was found. Values for $N_{CCN}^{A,NSD}$ are somewhat larger than $N_{CCN}$ derived from the three other instruments, which can be seen exemplarily in the slope of 1.15 and a $R^2$ of 0.92 for the fit presented in panel C of Figure 5. When comparing $N_{total}$ measured on ACTOS to $N_{total}$ derived from the NSDs measured on ACTOS (not shown), it was found that the latter was larger by a factor of 1.04 ($\pm$ 0.06), pointing towards a possible slight overestimation of number concentrations measured for NSDs on ACTOS. But overall, as the deviation between $N_{CCN}^{A,NSD}$ and the other data on $N_{CCN}$ is still close to 10%, a value which often denotes the uncertainty in aerosol particle counting instrumentation, we will not discuss this further but state that all methods used to derive $N_{CCN}$ gave similar results.

The above presented analysis helps to corroborate two points:

1) The use of an average $\kappa$, together with time-resolved NSDs, seems to be sufficient to retrieve $N_{CCN}$ for the marine aerosol examined in the present study. However, while a size dependent trend for $\kappa$ was not found for the here presented data, which enables the here made claim, seasonal changes in $\kappa$ are to be expected. It should also be added here that values for $N_{CCN}$ differ for different supersaturations. The supersaturation of 0.26% chosen for the work presented here is similar to that efficiently active in the examined trade wind cumuli. Therefore, the range of $N_{CCN}$ reported herein represents values valid for cloud droplet activation in the respective clouds.

2) The general agreement between $N_{CCN}$ derived at Ragged Point and on ACTOS again affirms that the ground based data measured for the present study can be assumed to be representative for the SCL and that the SCL generally was well mixed. This is also corroborated by the excellent agreement between wind speed measured at Ragged Point and on ACTOS as shown in Figure 6.

On grounds of these two points mentioned here, $N_{total}$ was derived from all NSDs measured at Ragged Point, and, as presented in the following, this rather extensive data-set was then used to discriminate between different air masses arriving at Ragged Point.

### 3.3 Particle number concentrations, aerosol types and their origin

5 #### 3.3.1 Particle number concentrations and corresponding different aerosol types

NSDs measured at Ragged Point which were used to derive $N_{CCN}$ in the previous section were also used to derive $N_{total}$. The derived values for $N_{total}$ represent particle number concentrations in the size range $\geq 25$ nm. The lower panels in Figure 7 show time series for $N_{total}$, depicted in either black, green or red (an explanation for the color coding follows below). For comparison, $N_{CCN}$ is shown again, as a grey line (using data derived from ground based NSDs, i.e. $N_{CCN}^{RP,NSD}$).

10 Now we will relate observed values of $N_{total}$ and $N_{CCN}$ to the already described three different types of NSDs that were observed in our campaigns, i.e., a discrimination between phases with different aerosol types is made based on measurements made at Ragged Point, meaning on altogether six weeks of data with a time resolution of roughly 30 minutes. Table 2 summarizes the characteristics of the resulting three different aerosol types.

During both months, there were long periods when $N_{total}$ and $N_{CCN}$ showed the same trends. At some times, however, 15 variations in $N_{total}$ were observed which were not seen for $N_{CCN}$ (see e.g., Figure 7 for DOY 324.5 to 329.5 and 107.5 to 111). During these times, $N_{total}$ was often at least twice as large as $N_{CCN}$. These increases in $N_{total}$ were caused by an increase in number concentrations for particles with sizes in the Aitken-mode, i.e., in a size range where the particles were still too small to act as CCN at the examined supersaturation. Corresponding NSDs typically were of the Aitken-type, as shown in the middle panel of Figure 1. The curve depicting $N_{total}$ in Figure 7 was colored green for phases where these Aitken-type NSDs were 20 present, or more precisely the curve was colored green when $N_{total}$ rose above 375 cm$^{-3}$ while $N_{total} \geq 2 \cdot N_{CCN}$. During these phases, $N_{CCN}$ can be seen to be mostly below 200 cm$^{-3}$.

For times of simultaneous trends in $N_{total}$ and $N_{CCN}$ and values for $N_{total}$ above 375 cm$^{-3}$, NSDs were generally of the accumulation-type (see right panel of Figure 1). The respective curve depicting $N_{total}$ was colored red. During these times $N_{total} < 2 * \cdot N_{CCN}$, and $N_{CCN}$ was above 200 cm$^{-3}$ in general.

25 For the remaining times, i.e., when the curve of $N_{total}$ in Figure 7 is shown in black, values for $N_{total}$ were below 375 cm$^{-3}$. $N_{CCN}$ mostly was below 200 cm$^{-3}$ during these times, and corresponding NSDs generally were of the marine-type (see left panel in Figure 1).

#### 3.3.2 Origin of the different aerosol types

In the following, we will discuss the influence of the origin of different air masses on the observed aerosol type. The connection 30 between both, air mass origin and aerosol type will be examined, based on either of the two, i.e., examining to which extent the origin determines the aerosol type and also to which extent the aerosol type is generally related to the air mass origin. We mainly focus on the extent to which aerosol arriving on Barbados was influenced by continental emissions from Africa,

and vice versa, how often elevated values of $N_{CCN}$ can be traced back to continental emissions from Africa. In this context, aerosol of the Aitken-type can be viewed as a sub-group of the marine-type aerosol, and therefore, for the inspection presented here, these two types were mostly summarized here.

The background of Figure 7 was colored based on an analysis of 10-day back-trajectories arriving at Ragged Point at a
height of 500 m, i.e., close to cloud base heights (*Siebert et al.*, 2013). This heights was chosen as being representative for the SCL, but also as trajectories calculated for 100 m (next lower elevation available) are more prone to uncerties. Trajectories were calculated every 4 hours with FLEXTRA (*Stohl et al.*, 1995) driven by operational ECMWF analysis/forecast fields with a horizontal resolution of 1° by 1° and 91 vertical levels. Whenever such a trajectory had crossed the Sahel or the Sahara in Africa, the background of Figure 7 was colored in grey or in orange stripes, respectively. More precisely, coloring of the
background was applied when the respective trajectory had crossed either one or both of the two following areas while being at a height of less than 3000 m at a time during the passage of these areas: 15° W to 0° W in all cases, 7° N to 15.5° N representing the Sahel region, and 15.5° N to 25° N representing the Sahara (as indicated by the grey and orange boxes in Figure 8). The grey box coincides with a region in the Sahel from which biomass burning emissions can be expected. The maps in the background of Figure 8 were taken from the NASA FIRMS Web Fire Mapper[1], indicating locations of fires for all
of November 2010 (left panel) and April 2011 (right panel), where small squares in white, light red or dark red indicate the presence of fire, larger numbers of fire events being indicating by darker coloring. In the Sahel region (roughly indicated by the grey box), fire was observed on every single day during both months, with an extension further eastward in November 2010. The orange box in Figure 8 represents a region with dust emissions, including the Western Sahara and Adrar Mountains for which frequent dust mobilization was observed in the past (*Schepanski et al.*, 2009). Note that the 10 days for which back-
trajectories were obtained do not allow to determine whether air masses had passed over the region east of the chosen areas. The height restriction of 3000 m was, in any case, on the lower end of the heights up to which dust or biomass-burning emissions can occur (*Prospero et al.*, 1981; *Smirnov et al.*, 2000; *Haywood et al.*, 2004), i.e., was chosen conservatively.

Additionally, trajectories shown in Figure 8 are colored in green, red or black, if they arrived at Ragged Point during a time when the aerosol was of the Aitken-, accumulation- or marine-type, respectively.
Looking at Figure 7, it can be seen that the redly colored phases marked by accumulation-type aerosol often coincide with times when the background was colored. This shows that air masses advected from Africa often lead to an increase in $N_{total}$ due to an increased particle number in the accumulation-mode, i.e., in the CCN size range. Of all trajectories (225 for the duration of both campaigns), those crossing one or both of the boxes shown in Figure 8 make up 24%, and for 68% of these, the aerosol was of the accumulation-type when arriving at Ragged Point. 76% of all 225 trajectories did not cross any of the two
boxes shown in Figure 8. Of these, 84% did not show an increase in particle number concentrations in the accumulation-mode. Table 3 summarizes these numbers and shows additional details.

As a side-note, it should be added that there was a short time of roughly a day at the beginning of the campaign in 2010 (during DOY 312), during which trajectories took a rather southern route and approached South America. Looking at Figure 8, these trajectories can be seen colored in red with starting points in the Atlantic, close to the equator. Some of the observed

---

[1]https://firms.modaps.eosdis.nasa.gov/firemap/

large numbers in $N_{total}$ during that time might also be attributed to particles originating from the South American continent, however, related air masses also might have had contact with Africa more than 10 days prior to arrival at Ragged Point.

Up to here, it was discussed how the origin of a trajectory from a certain region influenced the observed aerosol type arriving at Ragged Point. The following paragraph mentions how the presence of the different aerosol types can be traced back to the air mass origin.

Overall, for the three different aerosol types defined earlier, 1976 separate NSD-measurements (i.e., also 1976 values for $N_{total}$ and for $N_{CCN}$) exist. For 75% of these separate NSDs, the aerosol showed Aitken- or marine-type characteristics, and for 90% of all those, the corresponding trajectories had not crossed the two marked regions in Africa (see Table 4). For 57% of the measurements made during phases with accumulation-type aerosol, the trajectories had crossed one or both of the two marked regions in Africa. NSDs with a pronounced Aitken-mode were always connected to air masses that did not come from Afrika, and they made up $\sim$ 18% of all NSDs. All related information is given in Table 4.

Additionally, Figure 9 shows the history of the altitudes of air parcels arriving at Ragged Point as determined by the trajectories. Generally, elevations below 100 m were not observed, and most air parcels originated at heights well above 1000 m 10 days prior to arrival at Ragged Point. Also, for all aerosol types, almost all air parcels were found below 1000 m during the day prior to arrival. *Siebert et al.* (2013) showed that $N_{total}$ was similar in the SCL and in the cloud layer (up to a height of 2000 m) during the CARRIBA campaigns. The cumulus clouds observed frequently during CARRIBA form in air lifted from the SCL, and this air is replaced from above, explaining this observed similarity of the aerosol in these two different layers. Altogether, this and our observations of agreement between aerosol parameters as observed on ACTOS and at Ragged Point allows the conclusion, that the SCL is well mixed, down to the heights at which aerosol was sampled at Ragged Point.

### 3.3.3 Discussion of the different aerosol types

Aerosol of the accumulation-type often occurred during times when increased amounts of insoluble material were analyzed on the daily filter samples taken at Ragged Point (see top panel of Figure 7, denoted as ash). The corresponding increase in $N_{total}$ and $N_{CCN}$ can hence be expected to originate from biomass burning or desert dust particles, i.e., from particles of continental origin. This is in agreement with the above discussed origin of the respective air masses in the Sahara or Sahel region in Africa. While particles advected from Africa are known to generally occur in layers (SAL), particles from these layers descend and increase particle number concentrations at ground and in the SCL. Our measurements indicate that they are distributed comparably homogeneously, as otherwise the agreements between ground based and airborne measurements described above should not have been seen so clearly.

Particularly high amounts of ash were detected during April 2011 on DOY 100 and 102 to 106. The high value of ash on DOY 100 was only observed for this filter sample, and might be due to contamination. Roughly during DOY 103 to 107 (i.e., the end of DOY 106), $N_{total}$ and $N_{CCN}$ also indicate accumulation-type aerosol, however with lesser intensity than could be expected from the amount of ash on the filter. *Siebert et al.* (2013) described that particularly strong Saharan dust layers were observed during this phase in heights up to 3 km, where this information was obtained from LIDAR measurements that were done on the East Coast of Barbados, operated by the Max Planck Institute for Meteorology in Hamburg, Germany.

The corresponding NSDs measured on ACTOS show a strikingly strong particle mode, extending from $\sim$ 250 nm to above 2 $\mu$m, appearing as a clearly visible shoulder. With values of $dN/dlogD_p$ of 450 to 520 cm$^{-3}$ (and 280 cm$^{-3}$) at 300 nm for measurements on DOY 104 and 105 (and 106), respectively, these shoulders are much more pronounced than they are for the NSD shown as typical for the accumulation-type in Figure 1. Such large particle number concentrations in this size range were not observed at other times. This is indicative for mineral dust particles from the Sahara with respective sizes being present during this time.

The time from DOY 107 till the end of 110 was characterized by aerosol of the Aitken-type based on $N_{total}$, $N_{CCN}$ and NSDs. However, an increased mass concentration of ash was observed around DOY 109 and 110. During these times, comparably low wind speeds prevailed. They were around or below 3 m s$^{-1}$, while they otherwise were above 5 m s$^{-1}$, and sometimes even above 10 m s$^{-1}$ (see Figure 6). Winds recorded at Ragged Point usually came from east to south east (108° ($\pm$ 24°) and 104° ($\pm$ 20°) during November 2010 and April 2011, respectively), but during DOY 109 to 110 they came from north-eastly up to north-north-easterly directions. At the same time, several small fires were observed in the North of Barbados. These conditions might have led to an influence of local pollution on the ground based measurements at Ragged Point, which was already shortly mentioned in *Siebert et al.* (2013). Hence, some of the elevated values observed for $N_{total}$, which indeed originated in particles in the Aitken-mode size range, might not have been due to new particle formation, but due to large amounts of small particles from fresh biomass burning aerosol, and might explain the increased mass concentrations of ash during this time. Except for the here discussed local influence on the aerosol observed at Ragged Point, a general correlation between wind direction and $N_{total}$ was not observed.

Times with aerosol of the Aitken-type generally are marked by an increase in small particles, and it has already been described in Sec. 3.1 that these were recently nucleated, less than $\approx$ 3 days prior to arriving at Ragged Point. The respective trajectories, colored in green in Figure 8, show that the related air masses came from northern directions and had spent some days over the Atlantic. Curves and bents in them might imply that they were included in deep pressure systems, related with precipitation, maybe lowering the amount of available particles of larger sizes, hence lowering the particle surface area (i.e., the condensational sink) available for chemical reactions, increasing the likelihood for new particle formation. SO$_2$, a gaseous precursor generally assumed to play a crucial role in new particle formation, might have been emitted into the air masses by oceanic plankton in the form of DMS (dimethyl sulfide). Otherwise it might have been picked up over the North American continent, where a number of the 10-day back-trajectories originated. Unfortunately, the hygroscopicity analysis presented in Sec. 3.2 does not allow to draw conclusions on the chemical nature of the Aitken mode particles observed in this study, as the smallest particle diameters for which $\kappa$ was obtained were around 50 nm. But based on the modeling results by *Korhonen et al.* (2008) and *Merikanto et al.* (2009), it can be assumed that the recently nucleated particles likely originated in the free troposphere, at least to a large fraction. Alternatively, they were formed in the vicinity of trade wind cumuli, as suggested by *Wehner et al.* (2015). Following *Merikanto et al.* (2009), these small particles will eventually grow sufficiently so that they add to keeping up $N_{CCN}$, to which they contribute roughly half of all particles in the marine aerosol (*Merikanto et al.*, 2009).

Summarizing, we found that the marine aerosol observed on Ragged Point (and likely in large marine areas upwind of Barbados and the Caribbean in general) was largely dominated by the origin of the air masses. This is likely true not only for the two months during which measurements were made, but in general. When air masses were advected from Africa, in roughly 2/3 of all cases an increase in $N_{total}$ above 375 cm$^{-3}$ was observed, caused by particles in the accumulation-mode size range which might have consisted of rather insoluble material (biomass burning or mineral dust). These particles were,

however, observed to act as CCN at supersaturations above 0.26%. This is in line with *Twohy et al.* (2009), who also found that dust arriving at the Cape Verde Islands had acted as CCN in trade wind cumuli. It also agrees with *Karydis et al.* (2011), who show that mineral dust particles contribute significantly to CCN number concentrations, particularly in outflow regions of deserts, which, as shown here, for the Sahara might expand at least as far as Barbados.

Not all air masses originating from Africa carried continental particles in such large numbers. However, when air masses

had not originated in Africa, increases in $N_{total}$ above 375 cm$^{-3}$ mostly were caused by small particles recently formed over the Atlantic and $N_{CCN}$ generally was below 200 cm$^{-3}$.

### 3.4 Particles originating from sea spray

In the following section, it will briefly be discussed whether sea spray particles contributed noticeably to $N_{total}$. Following this brief excursion, we will examine the correlation between particles observed by the PDI during ACTOS flights in the SCL

in the size range from 500 nm up to cloud droplet sizes and particulate mass from sea spray aerosol as determined from filter samples taken at Ragged Point.

Regarding the first issue, Figure 10 shows the relation between $N_{total}$ to the horizontal wind speed as measured on top of the mast at Ragged Point. Wind speed data had been measured with a time resolution of one minute, and 30 minute averages were used for this comparison. No correlation between wind speed and $N_{total}$ was found. This indicates that the majority of the

particles observed at Ragged Point, and hence in the examined SCL did not originate from sea spray. This is in agreement with the earlier reported fraction of only 4 to 10% that particles in the observed sea spray mode contribute to $N_{total}$ (see Sec. 3.1). It further corroborates a statement made in Section 3.2, namely that the derived values for $\kappa$ indicate the presence of sulfates in the examined CCN size range up to 200 nm, rather than that of salts from sea spray.

Concerning the super-micron size range, Figure 11 shows the mass concentrations of Na$^+$+Cl$^-$ derived from the filter

samples taken on top of the mast at Ragged Point, in relation to different other parameters. On the filters, which sampled particles up into the super-micron size range, there was usually some amount of Cl$^-$ exceeding the stoichiometric relation between Cl$^-$ and Na$^+$. This is not surprising as NaCl is not the only salt present in seawater. Cl$^-$ and Na$^+$ are by far the most abundant ions. On the filters, they accounted for $\approx$ 80% of the total dissolved mass and around 60% of the total mass on average. In seawater, also Mg$^+$, SO$_4^{2-}$, Ca$^{2+}$ and K$^+$ are present, to name only the next most abundant ones. For CARRIBA,

there was no data available on the amount of Mg$^+$. SO$_4^{2-}$ made up close to 10% of the total mass, the sum of Ca$^{2+}$ and K$^+$ found on the filters was about 3% ($\pm$ 1%) of the total sampled mass. SO$_4^{2-}$ can be attributed either to particulate matter originating from seawater or to non-sea-salt sulfate. Hence we will take the sum of Na$^+$ and Cl$^-$ as a proxy for particulate

matter originating from sea spray. If we had used the stoichiometric amount of NaCl for the following comparisons, instead, the results we obtained would have been similar to those presented below.

In panel A of Figure 11, the mass concentration of $Na^+$+$Cl^-$ is related to the daily averaged wind speed (averaged over the time during which a filter was taken, where filters were changed daily around sunrise on Barbados, at 9:30 UTC). Grey and red data points represent data from November 2010 and April 2011, respectively, open symbols denote days during which

data from PDI is available. Overall, there might be a slight tendency for larger amounts of $Na^+$+$Cl^-$ being present for times with larger wind speeds, but this is not a clear trend. This might be attributed to comparably low wind speeds prevailing during both CARRIBA campaigns, as earlier studies reported that sea spray particles were only observed for wind speeds exceeding $10 \text{ m s}^{-1}$ (*Berg et al.*, 1998; *Massling et al.*, 2003), increasing in numbers with increasing wind speed (*O'Dowd and de Leeuw*, 2007; *Modini et al.*, 2015).

In panel B and C of Figure 11, the mass concentration of $Na^+$+$Cl^-$ is related to number and mass concentrations of large atmospheric aerosol particles which were observed with the PDI on ACTOS. The PDI was applied to measure droplet size distributions in clouds. However, it often also detected signals outside of clouds and in the SCL (where certainly no clouds were present), albeit less frequently than in clouds. These signals were interpreted as originating from hygroscopically grown particles in the super-micron size range. When comparing the mass concentration of $Na^+$+$Cl^-$ derived from the filter samples

to the number concentration of particles detected by the PDI in the SCL (panel C of Figure 11), a good correlation is seen. However, it should be mentioned here that PDI flights were only made for a small range of different atmospheric conditions. Diameters detected by the PDI were obtained at atmospheric conditions, and the respective dry particle sizes were derived as follows: We assumed that the particles originated from sea spray, therefore attributing them a $\kappa$ of 1, based on values derived in *Wex et al.* (2010a) and *Wex et al.* (2010b). From this, hygroscopic growth factors were calculated for ambient RHs. This was

done separately for each second during which particles were detected and the measured sizes were converted to dry sizes using these growth factors. The corresponding dry particle diameters were on average around 3.5 $\mu$m for all flights. Sub-micron particles with diameters down to around 500 nm were occasionally detected, but the majority of the counts detected with the PDI were found in the diameter range $> 1.5$ $\mu$m. For the further data evaluation, dry particle diameters together with the respective number concentrations were used to calculate the average particulate mass concentration detected by the PDI in

the SCL for each flight, assuming a density of 2.16 g cm$^{-3}$ (i.e., that of NaCl). Results are shown in panel C of Figure 11, together with a linear fit (forced through the origin), where the fit has a slope of 1.60, and $R^2$ is 0.42. Hence, particulate mass concentration derived from the PDI measurements is on average a factor of 1.60 above that of $Na^+$+$Cl^-$ sampled on the filters.

A direct comparison of the mass concentrations sampled on the filters with those derived from PDI measurements is, however, difficult for several reasons:

- As briefly touched upon above, we only used $Na^+$+$Cl^-$ from the filter samples in this comparison, and due to neglecting other ions, the mass concentration related to particles from sea spray derived from the filter samples might be up to 20% too low.

- Values for $\kappa$ and the density which were needed to derive mass concentrations from the PDI measurements were only estimates. The value for $\kappa$ was chosen slightly on the larger side of likely values (a range from roughly 0.85 to 1.1 is given in

*Wex et al.* (2010b) for the most hygroscopic mode in marine aerosol). Changing $\kappa$ to lower or larger values would increase or decrease mass concentrations derived for the PDI, respectively. An increase in $\kappa$ to that of NaCl (1.3) would lower the mass concentrations derived from PDI such, that they still exceed the filter values, but only by a factor of 1.3. However, such a large value of $\kappa$ for sea salt particles, which consist of a mixture of different salts, is not to be expected. Summarizing, using different, but still justifiable, values for $\kappa$ and the density would not erase the discrepancy.

- While filter data were daily averages, data from PDI were taken for the duration of an ACTOS flight, i.e. for 2 h at most. But meteorological conditions are overall comparably stable, and the mass derived from PDI measurements exceeds that determined from the filter samples in all cases, i.e., a constant bias is observed. This contradicts a randomized sampling error, therefore the differing sampling times can also not explain the discrepancy well, unless a daily cycle in particulate mass would be assumed to exist, for which there is no indication.

- The PDI has a lower cut-off in the size measurement of 1 $\mu$m (at ambient RH), while no clear lower cut-off for sizes of particles from sea spray can be given. Considering that sea spray particles form the observed third particle mode (see Sec. 3.1), the use of PDI data for the analysis presented here should result in too low values, where, however, those particles with sizes $< 500$ nm that were neglected will only contribute the smaller fraction to the overall particulate mass, due to their smaller sizes. Nevertheless, also this argument would rather support that the mass concentrations of $Na^+ + Cl^-$ derived from the filters would exceed those derived from PDI measurements, as the PDI misses small particles, which is not observed.

- Larger mineral dust particles could have been detected by the PDI, too. However, this is unlikely, as those days during which PDI data were taken (November 2010, from DOY 324 on), belong to phases with marine- or Aitken-type aerosol, except for one.

- Possibly some few large particles with sizes well in the super-micron size range were detected by the PDI but missed by the filter sampling, which would lead to mass concentrations from the PDI measurements exceeding those from $Na^+ + Cl^-$ sampled on the filters.

Although there are these issues concerning the here presented comparison, it might still be said that the correlation between the mass concentrations of $Na^+ + Cl^-$ derived from filter samples to those derived from PDI measurements shown in panel C of Figure 11 indicates that sea spray particles were present on Barbados and were found predominantly for larger sizes ($> 500$ nm) with a large contribution to sea spray related particulate mass from super-micron particles, as supposed already earlier in this study. As particles $< 500$ nm were not considered here, number concentrations of particles from sea spray will be larger than those presented in panel B of Figure 11, making up some percent of $N_{total}$ (see Sec 3.1). But based on the PDI measurements outside of clouds it was found that a small number of large particles (well up into the super-micron size range) was always present during the measurements, possibly contributing most of the sea salt aerosol mass derived from filter measurements. These few large sea spray particles might act as giant nuclei and hence could influence the formation of precipitation. But this topic is beyond the scope of the here presented work and will not be discussed further.

# 4 Summary and Conclusions

Summarizing, the here presented data-set consists of air-borne measurements and of six weeks (three weeks each during November 2010 and April 2011) of continuous in-situ aerosol data sampled in the Caribbean. Comparison of ground based and airborne in-situ aerosol measurements showed, that ground based measurements at Ragged Point were representative for the marine aerosol in the sub cloud layer arriving at Barbados. We deduce this from the similarity of NSDs measured on ACTOS
and at Ragged point, and also from the similar values of $N_{CCN}$ in the data-sets obtained on ground and airborne.

Based on these results, the continuous ground based measurements were used for a more thorough analysis of the aerosol and its origin. Three distinct types of air masses were observed, discriminated based on $N_{total}$, $N_{CCN}$ and the shape of the NSDs (see Figure 1 and Table 2). Accumulation-type aerosol was connected to long range transport from the Sahara and Sahel region, where either mineral dust from the desert and/or biomass-burning particles from the Sahel were included in raising
the particle number concentrations in the accumulation-mode. Although particles from both biomass burning and mineral dust are sometimes assumed to be of rather insoluble nature, they have been shown to contribute to CCN (*Engelhart et al.*, 2012; *Twohy et al.*, 2009; *Karydis et al.*, 2011), and an increase in $N_{CCN}$ due to their presence was also observed in the present study.

However, air masses from Africa not always contained increased particle loadings. Roughly 1/3 of the times when air
masses were advected from Africa, the aerosol was characterized as marine-type. In total, 75% of the time the aerosols arriving at Ragged Point were characterized as Aitken- and marine-type, and these aerosols mostly could be attributed to air masses which had not originated in Africa (to 90%). It is evident that sometimes new particle formation occurred in these aerosols during transport, an observation which was made during $\sim 18\%$ of the time. Particularly due to these periods, $N_{total}$ is not a good proxy for $N_{CCN}$, as then the former was larger than the latter by more than a factor of 2.
It can be speculated that there is a background of $N_{CCN}$ of some ten up to 200 cm$^{-3}$ constantly fed by particles formed by new particle formation which grew sufficiently large by condensational growth to become activated after a few days. This background $N_{CCN}$ level is higher when continental particles from Africa (and possibly rarely also from South America) are transported across the Atlantic. It should be mentioned, that $N_{CCN}$ depends on the supersaturation at which it was determined. But values of $N_{CCN}$ discussed in this study were taken at a constant supersaturation of 0.26%, and the Hoppel-minimum was
found to correspond to a supersaturation of 0.2 to 0.25%. Therefore, on average, $N_{CCN}$ as reported here can be taken to be relevant for droplet activation processes observed in the atmosphere in the marine environment of the Caribbean and possibly beyond. A companion paper examines in more detail, which conditions were found in separate clouds, concerning maxium supersaturation and number of activated particles (Ditas et al., in preparation).

Filter samples were taken, however, without any size segregation. They showed, that the majority of the total particulate mass
was comprised of Na$^+$ and Cl$^-$. A PDI had measured small droplets in the lower super-micron size range outside of clouds, which can be interpreted as haze particles formed on giant nuclei. Mass concentrations of these droplets were correlated with mass concentrations of Na$^+$+Cl$^-$ determined from the filter samples, and also a weak correlation with the wind speed was seen.

A correlation of $N_{total}$ to wind speed was, however, not observed, which is in agreement with the fact that particles originating from sea spray were found to only contribute a few percent to $N_{total}$ in accordance also with results by *Modini et al.* (2015). The data-set for the giant nuclei obtained from PDI measurements is small, but it might still be assumed that most of the mass of $Na^+$+$Cl^-$ found on the filters originated from sea spray particles in the super-micron size range. This is in agreement with e.g., *Bates et al.* (1998), *Glantz et al.* (2004), *Textor et al.* (2006), *Dunne et al.* (2014) and *Modini et al.* (2015), i.e., sea

spray particles might contribute much to particulate mass but do not contribute significantly to number concentrations of CCN, at least not for wind speeds as those prevailing during the CARRIBA campaigns, i.e., below $\approx 10$ m s$^{-1}$.

An average value for particle hygroscopicity of $\kappa = 0.66$ was found based on size segregated CCN measurements (0.56 during a phase of intense dust advection, 0.68 otherwise). *Pringle et al.* (2010) modeled $\kappa$ for surface locations in the Caribbean, obtaining values of 0.65 and 0.5 for April and November, respectively, however, with a seasonal cycle featuring lower values

during summer month. Our values and those of *Pringle et al.* (2010) agree within measurement uncertainty. A $\kappa$ of 0.66 suggests that sulfates were a major component of particles in the size range from $\approx 50$ to 200 nm, for which $\kappa$ was determined, and it points towards no or only a small fraction of organic compounds. This is in agreement with results reported by *Peter et al.* (2008) and *Korhonen et al.* (2008). However, measurements made at the same location on Barbados by *Kristensen et al.* (2016) in June and July 2013 derived much lower values for $\kappa$ of 0.2 to 0.5, again in agreement with *Pringle et al.* (2010), indicating a

possibly larger organic fraction in the particulate matter at that time of the year. Coming back to the filter measurements, it has to be argued that an estimation of the particle hygroscopicity $\kappa$ from the chemical composition found on the filters would yield too large values, due to the predominance of particulate mass contributed by a few large and very hygroscopic particles. It is worth noting that also *Juranyi et al.* (2010) and *Silvergren et al.* (2014) found that $\kappa$ derived based on chemical composition exceeded that derived from CCN measurements, for remote continental and Arctic aerosol, respectively.

The average $\kappa$ derived from CCN measurements could be used, together with measured NSDs, to calculate $N_{CCN}$ with high accuracy. Also in *Juranyi et al.* (2011), $N_{CCN}$ could be modeled well for a time series of 17 month for the alpine site of Jungfraujoch in Switzerland, based on one average value for $\kappa$ together with time and size resolved NSD measurements. Our observations together with these from literature, and the fact that $\kappa$ from filters has been observed to exceed that derived from CCN measurements, might indicate that a derivation of $\kappa$ based on CCN measurements might be advantageous compared to

using the chemical composition from filter samples when aiming at the determination of $N_{CCN}$ for further use in modeling.

Summarizing, the here presented results corroborated and extended earlier knowledge about the marine aerosol in the Caribbean. Although the aerosol on Barbados often is considered clean marine, still continental aerosol advected mainly from Africa can influence $N_{total}$, $N_{CCN}$ and the shape of the NSDs, carrying comparably high loadings of mineral dust or biomass burning particles. Therefore, a direct anthropogenic influence on the marine aerosol in the Caribbean, brought on by

man-made fires, can not be completely excluded. This is in line with results reported in *Hamilton et al.* (2014), where it is described that pristine aerosols (i.e., those free from anthropogenic influence) on Earth are largely only found on the Southern Hemisphere, while these pristine environments are only found temporarily and spatially patchy on the Northern Hemisphere. Information on particle composition derived from our data suggests that the majority of particles in the CCN size is made up of sulfates, where these particles mostly originated from new particle formation. An influence of organic substances on particle

hygroscopicity in the CCN size range could not be seen for the month during which measurements were made, but might be present in summer (*Kristensen et al.*, 2016). Few large sea salt particles contribute the major fraction of total particulate mass but only contribute a small amount to the total particle number concentrations. Studies like the present one can help completing the picture concerning atmospheric aerosol properties, can be used as a base for model studies or can help corroborate results obtained in model studies, strengthening our trust in the respective models.

5 **Appendix A: Accounting for particle losses**

**A1   Corrections due to the reduced flow rate used for CPC measurements and due to inlet losses**

At Ragged Point, particle number concentrations were measured with a CPC which was fed with an aerosol flow rate of $0.5 \, \mathrm{l \, min^{-1}}$ instead of its regular flow rate of $1 \, \mathrm{l \, min^{-1}}$, which already necessitated that all measured values be multiplied by a factor of 2. It was determined at the home laboratory that the reduced flow rate furthermore resulted in a reduction of the 10   counting efficiency, such that measured values additionally had to be corrected by a factor of 1.2.

Losses in all inlet tubes, from the top of the Ragged Point tower down to the inlet of the different instruments, were calculated based on length and diameter of the inlet tubes and the flow passing through them. This resulted in a size dependent correction factor ranging from around 1.135 for number concentrations for the smallest measured particle size (25 nm) up to below 1.005 for the largest measured sizes at 500 nm.

15   Size dependent particle losses in the two nafion and the one diffusion dryer had all been determined through measurements in the laboratory. Particularly large losses were found for particles with smaller sizes in the diffusion dryer (e.g., correction factors of 1.03, 1.46 and 2.11 had to be applied at 500, 100 and 50 nm, respectively).

All of the here described corrections were obtained based on calibration measurements in the laboratory, and on the calculation of size dependent losses in tubes which can be calculated precisely, too. The size dependent correction factors increases 20   with decreasing particle size, so that the overall correction factor from the here described losses increases from 2.5 to 3 over the size range from 500 to 200 nm, while it is, e.g., above a factor of 4 below 70 nm and up to a factor of 7 for the lowest measured sizes. Hence, the uncertainty for the correction factors are larger for smaller particle sizes. This might contribute to an overestimation of the particle number concentrations obtained in the size range below the Hoppel minimum, as observed when NSDs measured at Ragged Point were compared to those measured on ACTOS (see Figure 1 and App. A3).

25   Nevertheless, applying the corrections was mandatory, and all correction factors described here were accounted for before NSDs measured at Ragged Point were used for further data evaluation.

**A2   Correcting data from the mini-CCNC with respect to inlet losses**

The mini-CCNC operated at Ragged Point also had to be corrected for particle losses, however, as it did not detect the smaller particles (roughly below the Hoppel minimum), the correction factor had to be determined separately. These total losses

30 originating from the inlet tubes and the dryers in the particle size range detected by the mini-CCNC were determined as follows:

All NSDs measured at Ragged Point during November 2010 and April 2011 were considered. The integrated particle number concentrations in the size range from a diameter ($D_p$) of 68 nm (Hoppel minimum) up to 500 nm were determined twice for each NSD, once based on data corrected following App. A1 ($(\frac{dN}{dlogD_p})_{corr}$), and once based on data which had not yet been

5 corrected for losses in the inlet tubes and in the dryers ($(\frac{dN}{dlogD_p})_{uncorr}$). The overall average factor $F_i$ (where $i$ stands for "losses in the inlet system") between these two sets of particle number concentrations was found to be 1.405 ($\pm$ 0.030). The following equation demonstrates the above described approach:

$$F_i \cdot \int_{68\ nm}^{500\ nm} (\tfrac{dN}{dlogD_p})_{uncorr} dlogD_p = \int_{68\ nm}^{500\ nm} (\tfrac{dN}{dlogD_p})_{corr} dlogD_p$$

This correction factor $F_i$ was applied to all values measured for $N_{CCN}$ at 0.26% supersaturation at Ragged Point before considering them in the comparisons presented in the above study.

## A3 Corrections necessary due to the use of a cyclone

As described in Section 2, a cyclone was installed in the inlet system at Ragged Point. Due to this cyclone, particle losses

occurred for sizes > 200 nm, and particles > 500 nm were completely removed. Here it will be described how data measured on ACTOS were used to determine the fraction of large particles that was not detected by the Ragged Point data-set.

In a first step, a comparison was made between all NSDs measured on ACTOS and the respective NSDs measured at Ragged Point. From the ACTOS data-set, for each flight one average NSD from all measurements in the SCL was used. From the ground based data-set, similarly all NSDs measured during the duration of each flight were averaged (all these NSDs had been

corrected as discussed in App. A1). To enable the comparison, measured particle number concentrations from both, Ragged Point ($N_{RP}$) and ACTOS ($N_{ACTOS}$), were interpolated so that data from both data-sets was available at the same diameters. The comparison resulted in the curves shown in Figure 12. NSDs measured at Ragged Point overestimate particle number concentrations for diameters below $\sim$ 70 nm by more than 10%, an observation for which a possible explanation was already given in App. A1. For particle sizes above $\sim$ 200 nm, the effect of the cyclone starts to be visible and at 300 nm, already 50%

of all particles are captured in the cyclone. The kink at about 300 nm likely originates in the fact that ACTOS measurements for smaller sizes were done with the SMPS while those above this size were made with the OPC, i.e., the transition from one instrument to an other caused a slight jump in the ACTOS data.

Figure 13 shows the efficiency of the cyclone as determined in the home laboratory (grey dots, and a grey line representing the corresponding fit), showing a cut-off diameter of the cyclone (i.e., where 50% of all particles are removed) of 525 nm.

This diameter relates to the humidified aerosol, e.g., as it enters the measurement container at Ragged Point. When assuming $\kappa = 0.66$ and relative humidities of 70%, 80% and 90% (corresponding to growth factors of 1.43, 1.6 or 1.945, respectively) the red lines in Figure 13 are obtained as the efficiency curves for the respective dried aerosol. Also shown in Figure 13 is the black curve from the upper panel (here now depicted as black squares), i.e., the average ratio of number concentrations measured at

Ragged Point to those measured at ACTOS ($N_{RP}/N_{ACTOS}$). This data fits well to the thickest red line, albeit being slightly broader. The relative humidities at Barbados were around 70% to 80% at temperatures of about 27°C in November 2010 and 65% to 75% at temperatures of about 25.5°C in April 2011. The temperature inside the measurement container was kept at around 23°C to 24°C, which indeed would lead to relative humidities of around 80% up to even above 90% for the aerosol in the tubing inside of the container. The corresponding dry cut-off diameter of the cyclone was $\approx$ 275 nm. In general, the

agreement for particle sizes $\geq$ 180 nm between the impactor efficiency curve determined in the laboratory and values obtained for $N_{RP}/N_{ACTOS}$ justifies the assumption that the latter describes the impaction efficiency well, and hence this curve could be used to correct NSDs measured at Ragged Point for losses occurring in the cyclone (for particle sizes $\geq$ 180 nm). However, this correction alone is not able to account for particle losses in size ranges where no particles passed the cyclone ($\geq$ 500 nm).

To also correct for losses occurring for particle sizes in the CCN size range $\geq$ 500 nm, a procedure comparable to the one

introduced in App. A2 was done next. NSDs from ACTOS were artificially lowered by multiplying them with the impaction efficiency, or, more precisely, with $N_{RP}/N_{ACTOS}$ in the size range $\geq$ 180 nm and with 0 for $\geq$ 500 nm. Then, integrated particle number concentrations were made from 68 nm on for both, the original NSDs and those artificially lowered. The factor between these values then yields the fraction of larger particles (i.e., those missing due to the cyclone but additionally also those > 500nm which were not detected at all at Ragged Point), compared to all particles.

The following equation again demonstrates the above described approach, i.e., how the factor $F_l$ (where $l$ stands for "missing larger particles") was obtained:

$$F_l \cdot \int_{68 \ nm}^{2500 \ nm} \left( \frac{dN}{dlogD_p} \frac{N_{RP}}{N_{ACTOS}} \right) dlogD_p = \int_{68 \ nm}^{2500 \ nm} \frac{dN}{dlogD_p} dlogD_p$$

$F_l$ was very similar for most cases: It was determined to be 1.15 ($\pm$ 0.05) on average for the measurements made in November 2010 and also for those in April 2011, with the exception of the time span from DOY 103 to 107, where the average $F_l$ was 1.26 ($\pm$ 0.01). This particular time span coincides with the time when also slightly lower $\kappa$-values were observed and during which a strong dust event occurred. The difference in $F_l$ during and outside of this dust phase was larger than the uncertainty in this value. Hence, a value of $F_l$ of 1.15 was used for all times except for the time span from DOY 103 to 107 for which 1.26

was used.

$F_l$ now is the factor by which the integrated particle number concentrations in the accumulation-mode (or, more precisely, in the size range $\geq$ 68 nm) have to be multiplied in order to correct for the above mentioned missing large particles in the Ragged Point data-set. This factor was used to correct $N_{CCN}$ obtained from the mini-CCNC at Ragged Point at a supersaturation of 0.26%, and the corrected values of $N_{CCN}$ can then be expected to be valid for the atmosphere. $F_l$ was also used to correct the

integrated particle number concentrations in the size range $\geq$ 68 nm when $N_{CCN}$ or $N_{total}$ were derived from NSDs measured at Ragged Point (without further correction for particles in the size range < 68 nm). The thus obtained values for $N_{CCN}$ and $N_{total}$ as determined from Ragged Point measurements can then also be expected to be representative for the atmosphere.

*Acknowledgements.* We thank Prof. em. Joe Prospero for access to his measurement station at Ragged Point. We are grateful to Dr. Stephan Henne, EMPA (Swiss Federal Laboratories for Materials Science and Technology) for kindly providing the 10-day back-trajectories (to be found at http://lagrange.empa.ch/FLEXTRA_browser). We acknowledge the use of FIRMS data and imagery from the Land Atmosphere Near-real time Capability for EOS (LANCE) system operated by the NASA/GSFC/Earth Science Data and Information System (ESDIS) with funding provided by NASA/HQ. This project was partly funded by DFG (SI 1534/3-1) and the European FP7 project BACCHUS

5   (Impact of Biogenic versus Anthropogenic emissions on Clouds and Climate: towards a Holistic UnderStanding, grant agreement no. 49 603445). Thanks also to David Farrel from the Caribbean Institute for Meteorology and Hydrology (CIMH) for logistical support.

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

**Table 1.** Measured and derived parameters and the respective instrumentation used at Ragged Point and on ACTOS.

| parameter | symbol / abbreviation | instrument / underlying data | measurement range |
|---|---|---|---|
| **Ragged Point** | | | |
| particle number size distribution | NSD | DMPS-system | 25 to 500 nm |
| total particle number concentration | $N_{total}$ | integrated NSD | - |
| CCN number concentration | $N_{CCN}^{RP,CCNC}$ | mini-CCNC | 0.26% supersaturation |
| | $N_{CCN}^{RP,NSD}$ | integrated NSD | - |
| particle hygroscopicity | $\kappa$ | CCNC with DMPS-system | at 0.07%, 0.1%, 0.2%, 0.4% and 0.7% supersaturation |
| **ACTOS** | | | |
| particle number size distribution | NSD | SMPS-system and OPC | 6 to 230 nm 250 to 2500 nm |
| total particle number concentration | $N_{total}$ | CPC | > 6 nm |
| CCN number concentration | $N_{CCN}^{A,CCNC}$ | mini-CCNC | 0.26% supersaturation |
| | $N_{CCN}^{A,NSD}$ | integrated NSD | - |
| cloud droplet size distributions | | PDI | 1 to 180 $\mu$m |

**Table 2.** Overview over the three observed distinct aerosol types. Typical NSDs are shown in Figure 1.

| | $N_{total}$ [cm$^{-3}$] | $N_{CCN}$ [cm$^{-3}$] | shape of NSD |
|---|---|---|---|
| A) marine-type | $< 375$ | $< \approx 200$ | clearly visible Hoppel minimum |
| B) Aitken-type | $> 375$ | $< \approx 200$ | pronounced Aitken-mode and clearly visible Hoppel minimum |
| C) accumulation-type | $> 375$ | $> \approx 200$ | pronounced accumulation-mode |

**Table 3.** Statistics on trajectories. (Percentages in brackets indicate, how many of those mentioned before the bracket belonged to the respective type.)

| | number of trajectories | from Africa[a] | not from Africa[b] |
|---|---|---|---|
| 2010 | 118 | 38% (64% accumulation-type) | 62% (73% Aitken- or marine-type) |
| 2011 | 107 | 7% (88% accumulation-type) | 93% (93% Aitken- or marine-type) |
| both years | 225 | 24% (68% accumulation-type) | 76% (84% Aitken- or marine-type) |

[a]: trajectory crossed one or both boxes representing the Sahara and the Sahel.

[b]: trajectory did not cross any of the two boxes representing the Sahara and the Sahel.

**Table 4.** Statistics on NSDs. (Percentages in brackets indicate, how many of those mentioned before the bracket had trajectories originating from the respective region.)

| | number of NSDs | marine- and Aitken-type[c] | accumulation-type[c] |
|---|---|---|---|
| 2010 | 817 | 44% and 16% (75% not from Africa)[b] | 40% (58% from Africa)[a] |
| 2011 | 979 | 67% and 20% (99% not from Africa)[b] | 13% (53% from Africa)[a] |
| both years | 1796 | 57% and 18% (90% not from Africa)[b] | 25% (57% from Africa)[a] |

[a]: trajectory arriving at the time of the NSD measurement crossed one or both boxes representing the Sahara and the Sahel.

[b]: trajectory arriving at the time of the NSD measurement did not cross any of the two boxes representing the Sahara and the Sahel.

[c]: Following the characteristics given in Table 2.

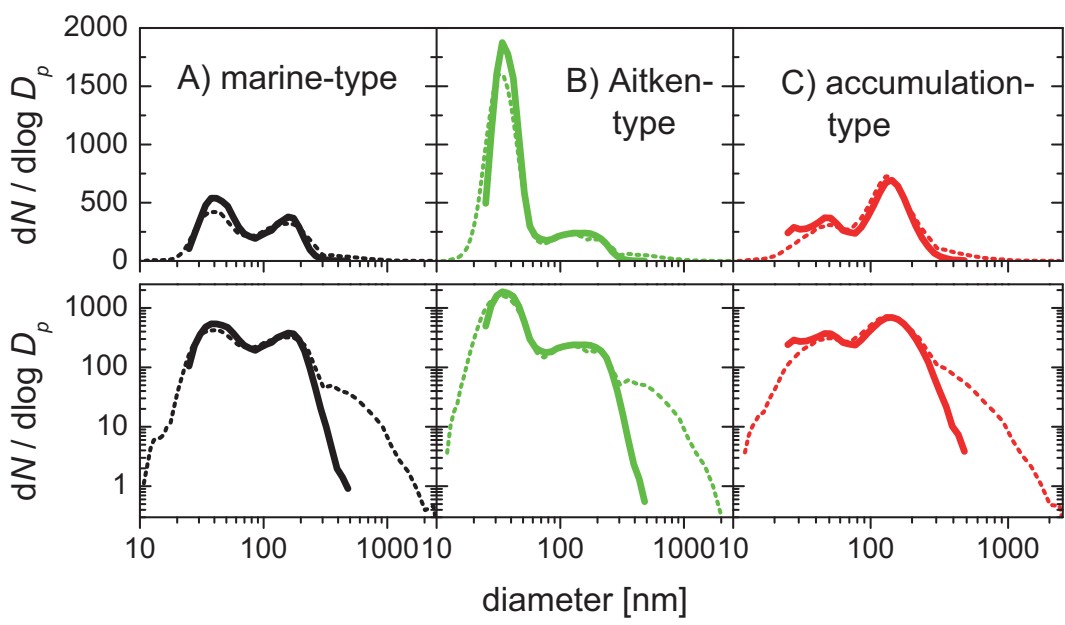

**Figure 1.** Exemplary NSDs measured during three different ACTOS flights in the SCL between 100 and 400 m on ACTOS (thin dotted lines) and on ground (thick solid lines), shown with a linear (top) and a logarithmic (bottom) scaling on the y-axis. Ground based measurements from 25 nm to 500 nm were done using a DMPS, while size distribution measurements on ACTOS were composed from measurements of a SMPS, measuring from 6 up to 230 nm and an OPC covering the size range from 250 nm to 2500 nm. All data shown here and on any other plot in this study are given for STP.

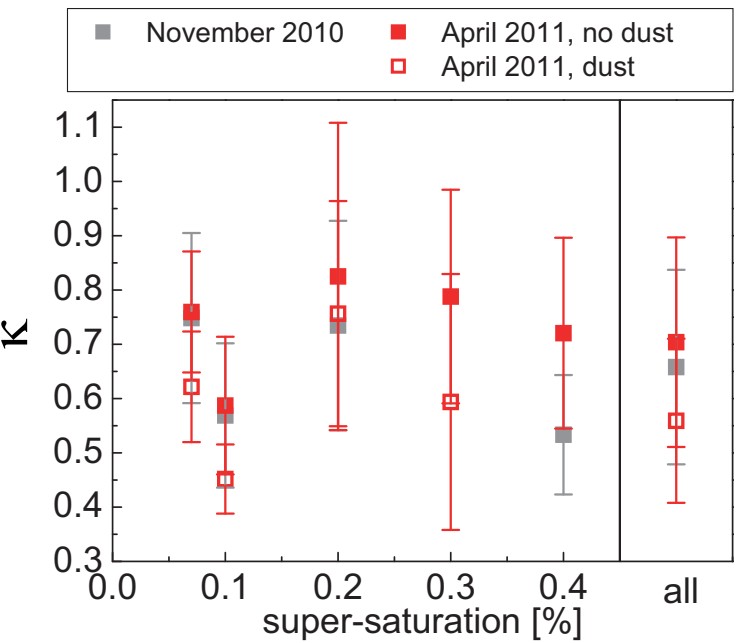

**Figure 2.** $\kappa$-values derived at different supersaturations for November 2010 and April 2011 (red open squares represent the time from DOY 103 to 107 when dust was observed). Error bars represent the standard deviation resulting from averaging all $\kappa$-values obtained at the respective supersaturations.

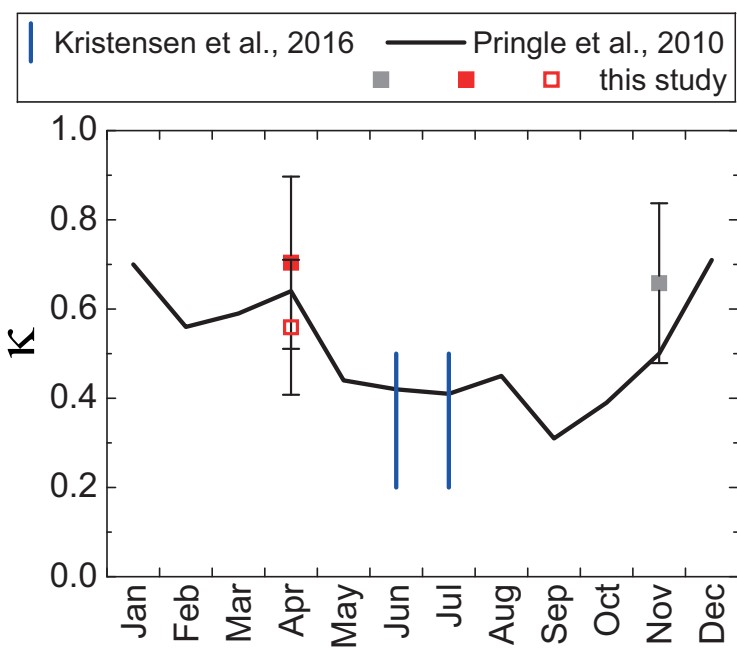

**Figure 3.** $\kappa$-values reported in literature based on modeling (*Pringle et al.*, 2010) and measurements (*Kristensen et al.*, 2016, data taken during SALTRACE in 2013) and derived in this study for November 2010 and April 2011 (color code for the latter similar to that used in Fig. 2).

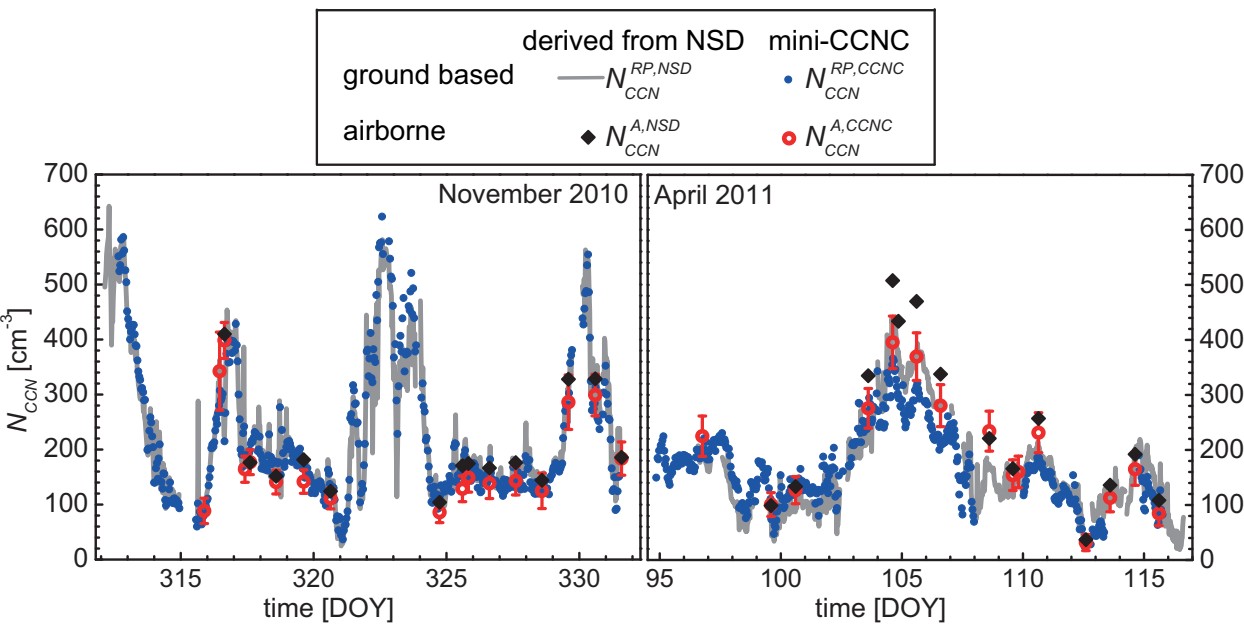

**Figure 4.** Time series of $N_{CCN}$ measured with the mini-CCNC at a supersaturation of 0.26% both on ground and airborne, and determined from NSDs for a cut-off diameter of 68 nm, corresponding to a $\kappa$ of 0.66 at 0.26% supersaturation.

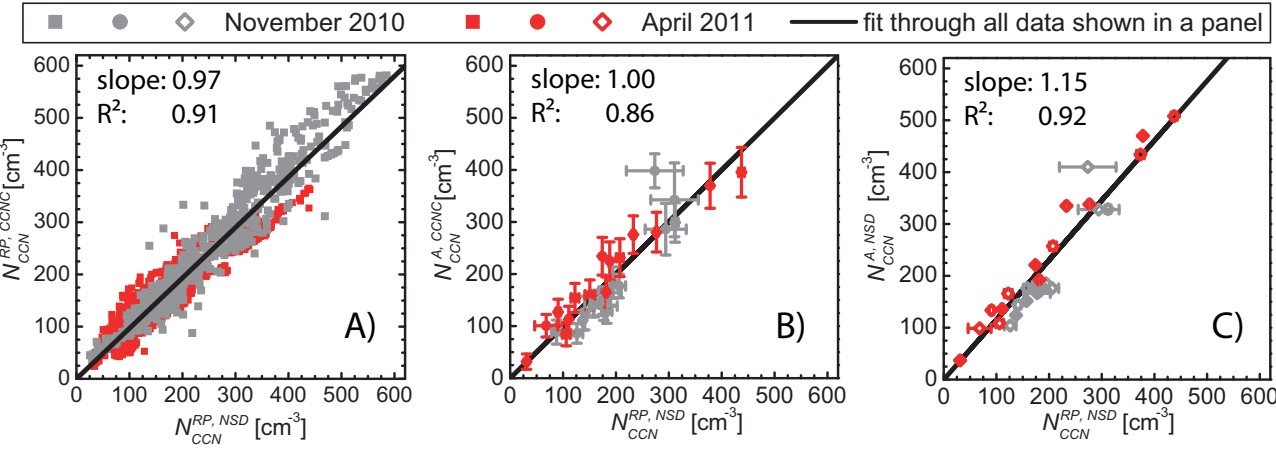

**Figure 5.** Scatterplot comparing the $N_{CCN}$ data-sets shown in Figure 4. $N_{CCN}^{RP,NSD}$ is compared to $N_{CCN}^{RP,CCNC}$, $N_{CCN}^{A,CCNC}$ and $N_{CCN}^{A,NSD}$ in the left, middle and right panel, respectively. Slope and $R^2$ for these fits is given in the panels. (For clarity, the display of error bars was omitted in the left panel, where, on average, a standard deviation of 20% was found for $N_{CCN}^{RP,CCNC}$.)

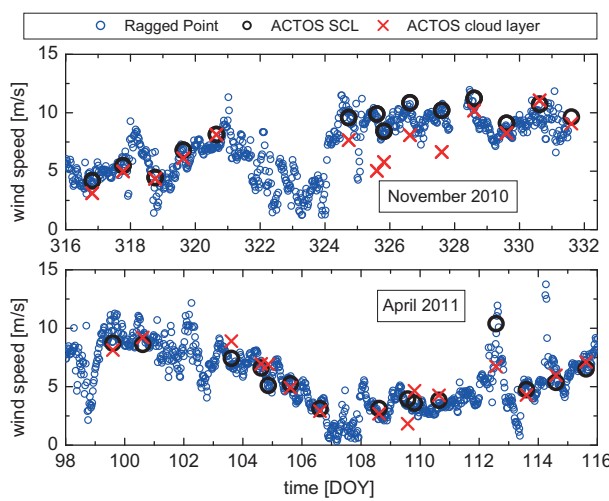

**Figure 6.** Wind speed as measured at Ragged Point (30 minute average) and on ACTOS (averaged for each research flight).

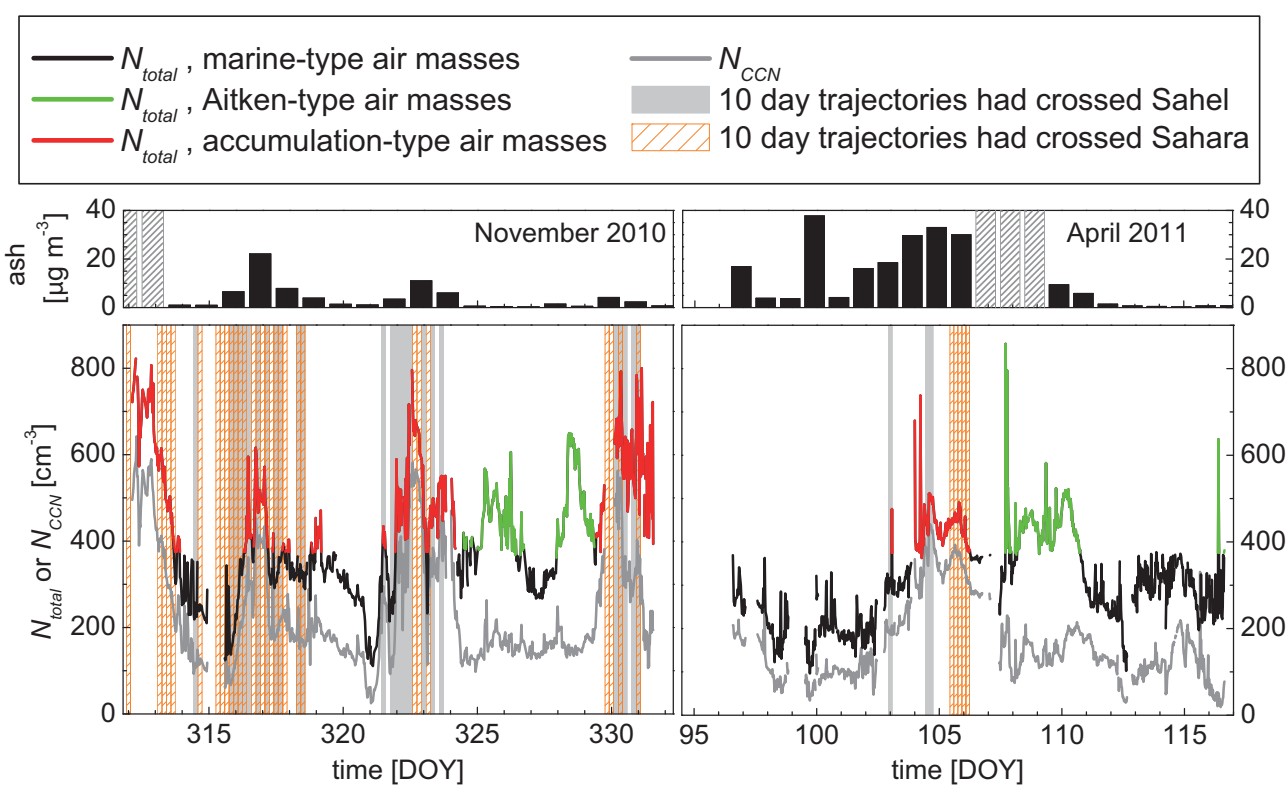

**Figure 7.** Time series. Upper panels: mass of the residuals from filter samples remaining after heat treatment (ash) - grey striped areas indicate days for which no data was available. Lower panels: $N_{CCN}$ (same as in Figure 4) and $N_{total}$, both derived from NSDs measured at Ragged Point. For the color coding of $N_{total}$ see the legend (based on the definitions of different air masses given in Table 2). The background is colored light grey or in orange stripes when trajectories indicated that air masses were advected from Africa (see the two boxes in Figure 8).

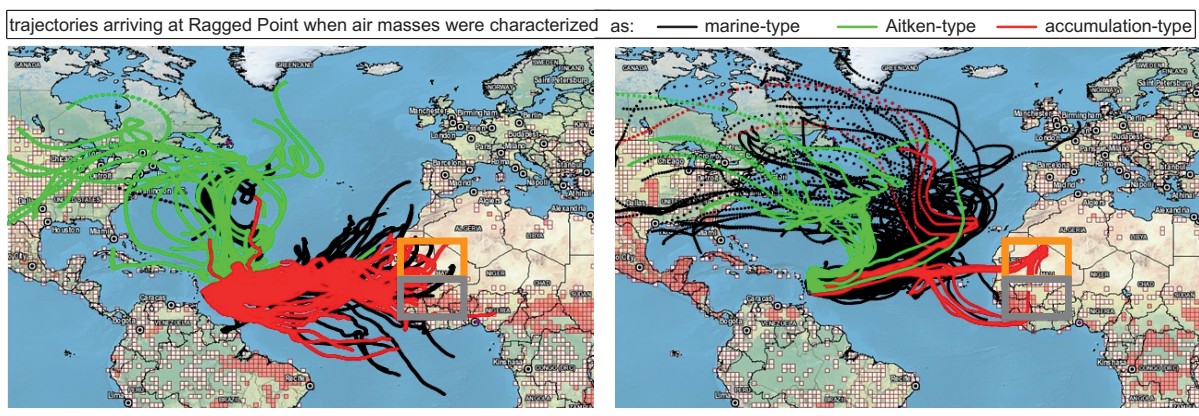

**Figure 8.** 10 day back-trajectories arriving at Ragged Point at a height of 500 m. Trajectories were calculated every 4 hours (each calculation is shown as a separate dot, which is separately visible when air masses moved respectively fast). The trajectories shown here are those arriving at Ragged Point during the measurements periods in November 2010 (left panel) and April 2011 (right panel). The map in the background was taken from the NASA FIRMS Web Fire Mapper. The orange and grey box represent those regions in the Sahara and Sahel, respectively, which a trajectory had to cross to be characterized as being influenced by either of them. More details are given in the text.

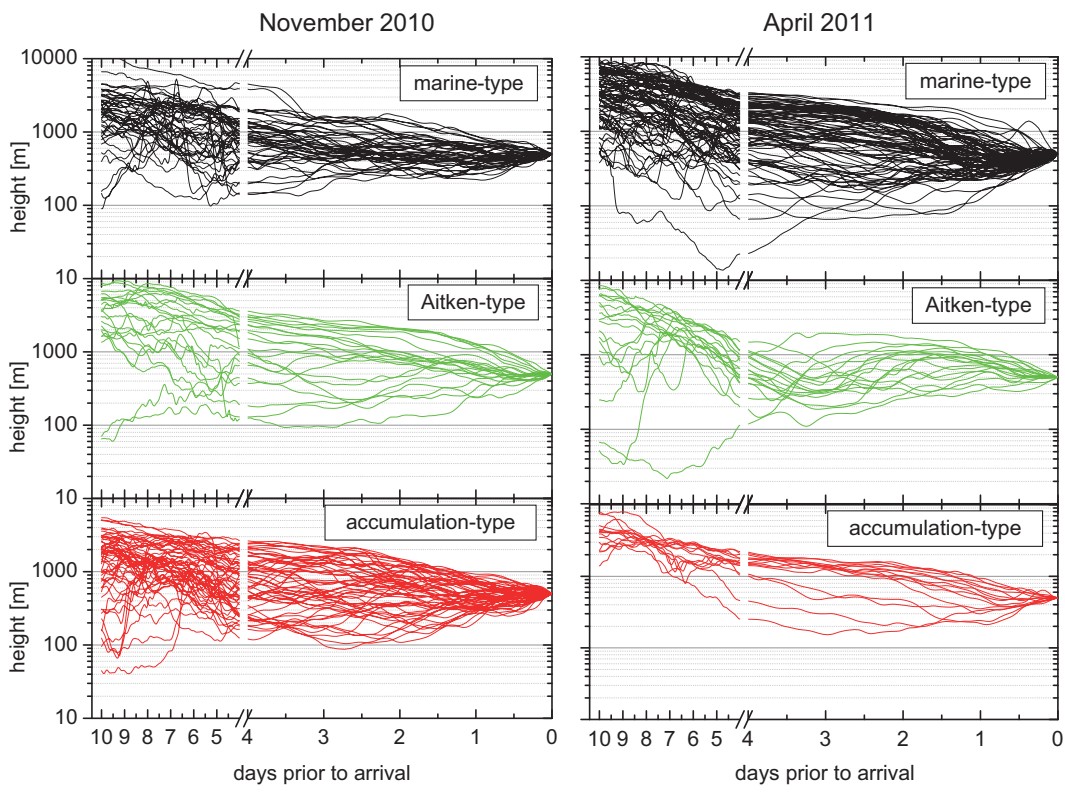

**Figure 9.** Heights of the 10 day back-trajectories prior to arriving at Ragged Point.

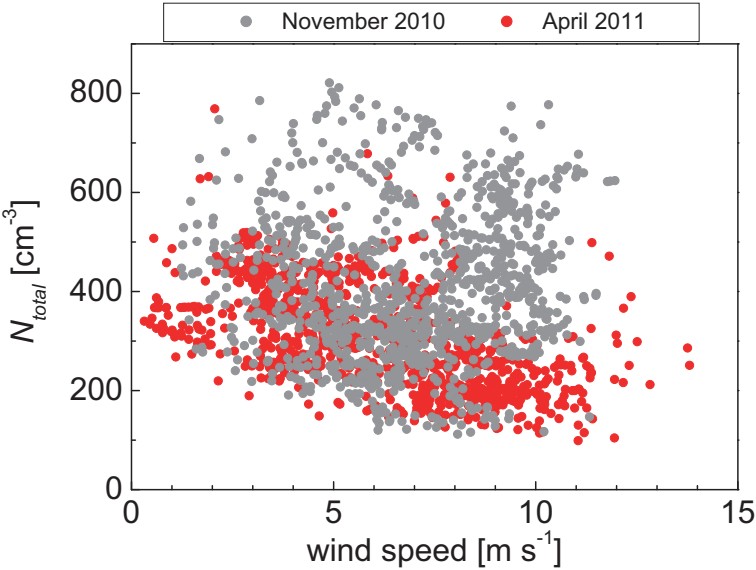

**Figure 10.** Comparison of horizontal wind speed to $N_{total}$.

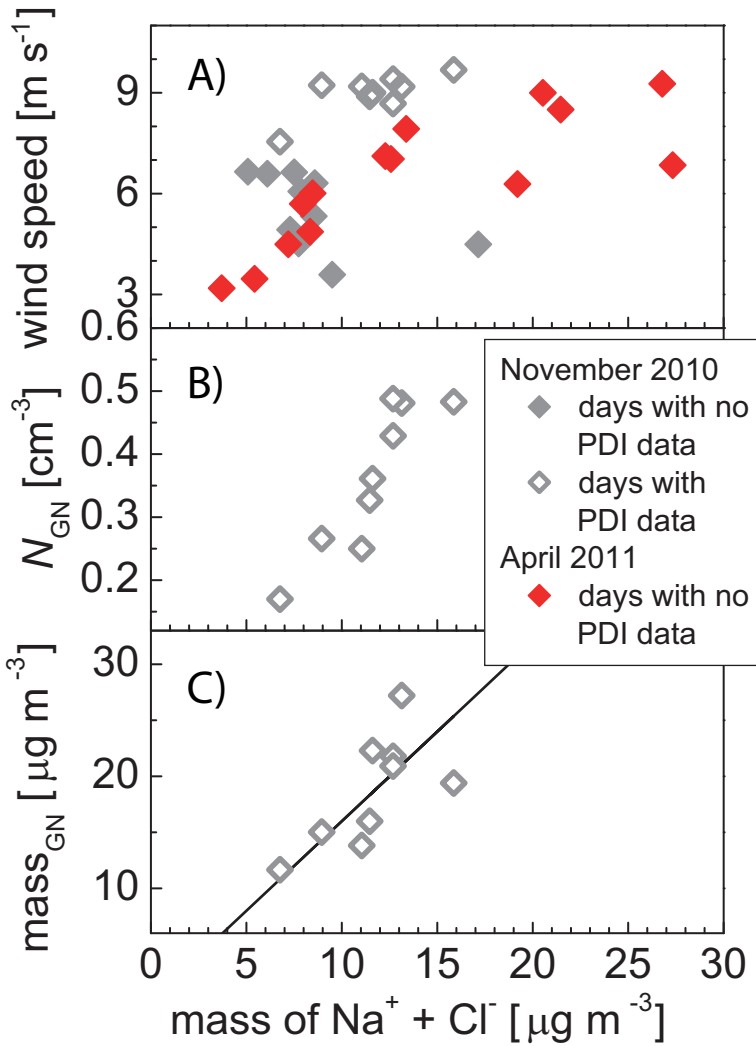

**Figure 11.** Mass concentrations of $Na^+$+$Cl^-$ from filter samples taken at Ragged Point, in comparison to wind speed (upper panel) and to number and mass concentrations (middle and lower panel, respectively) of particles detected with the PDI in the SCL.

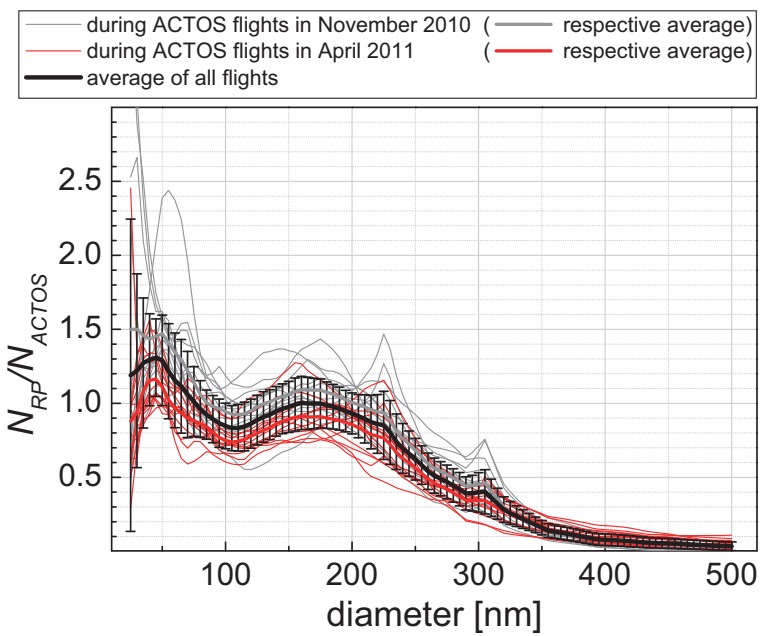

**Figure 12.** Bin-wise comparison of NSDs measured at Ragged Point and on ACTOS (i.e., $N_{RP}/N_{ACTOS}$).

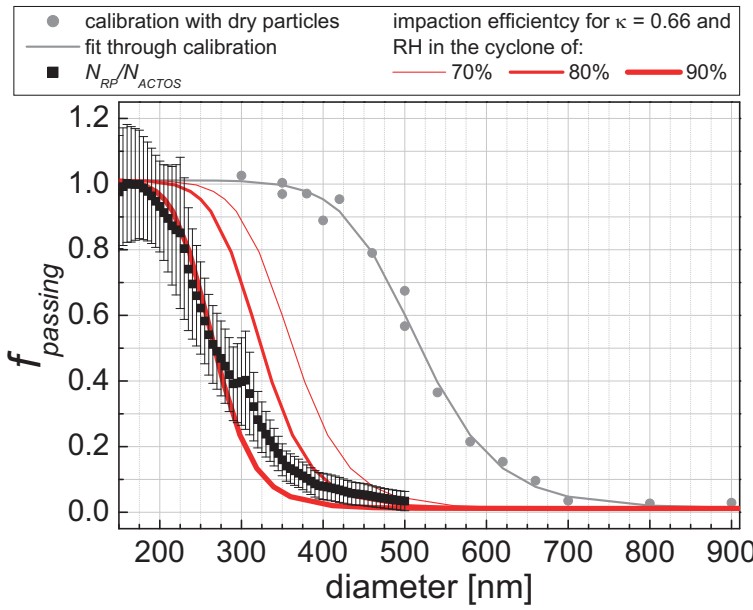

**Figure 13.** Impaction efficiency of the cyclone, i.e., the size dependent fraction of particles passing the cyclone ($f_{passing}$). The grey dots result from calibration measurements done in the home laboratory with dry size selected ammonium sulfate particles, the grey line is the respective fit. The red lines originate from this fit when it is assumed that particles were hygroscopically grown while passing the cyclone and were dried only afterwards. Data shown in black is the same as in Figure 12, i.e., $N_{RP}/N_{ACTOS}$.