# Peer review of "Aerosol arriving on the Caribbean island of Barbados: Physical properties and origin"

_Atmospheric Chemistry and Physics, 2016_

## Referee Comment (RC1) · Anonymous Referee #1 · 2 May 2016

This manuscript investigated the aerosol arriving at Barbados and found 3 typical number size distributions (NSD), classified as marine-type, Aitken-type and accumulation-type. Size distributions on ground where compared with airborne measurements. Hygroscopicty and CCN number concentrations were investigated, also in regards of their origin. Sea spray was found to be a minor source of CCN. The manuscript is well written and structured. Measurement uncertainties and data quality are well documented and possible limitations and alternative interpretations are discussed. I recommend publication, after minor revision.

**General Comments:**

1. The reader would strongly benefit, if all Figures had a legend. Figures 2, 5, 6, 7, 8 do not have a legend, while others have.

2. The discussion of the particle hygroscopicity is very informative, however, I did not find a comparison of the different kappa values for the different NSD types. Did the different types show different or similar kappa-values? Did all Aitken/Nucleation mode types show a similarly high kappa, or was there a variation indicating different nucleation precursor? Also how representative is the kappa value for these Aitken mode types, as the maximum in the NSD is below the critical diameter. Can these kappa values be used to get an indication how these particles were formed, in terms of aerosol precursors.

**Specific Comments**:

**Page 3 Lines 17ff.:** It is noted that nucleation in the free troposphere is derived from DMS-derived $H_2SO_4$. Kirkby et al. showed that pure binary $H_2SO_4$-$H_2O$ nucleation cannot explain ambient formation rates and that it requires ternary $NH_3$-$H_2SO_4$-$H_2O$ nucleation even in the free troposphere. This is nowhere mentioned or discussed. These findings should at least be mentioned.

**Page 5 Line 8:** How long were the filters stored and at which temperature before measurement. Could this influence the results?

**Page 5 Line 19:** What is the absolute (wet) cut-off of the cyclone, were all droplets removed?

**Page 5 Line 29:** Which neutralizer was used, Krypton, X-ray? Which charging efficiency was applied?

**Page 10 Lines 26ff.:** During Aitken-type events $N_{total}$ is often twice as high as $N_{CCN}$. What does this imply on the importance of new particle formation to CCN production? Previous modelling studies (e.g. Merikanto et al., 2009) suggested that up to 45

**Page 11 Section 3.3.2:** The authors suggest that Aitken-mode NSD might have entrained from the free troposphere, as also previously suggested by several studies. Did you find any indication of this in FLEXTRA? Additionally, FLEXPART dispersion modelling (https://www.flexpart.eu/) would have been my choice instead of its predecessor FLEXTRA. Why was it not used?

**Page 27 Table 3 and Table 4.** I find these two tables somewhat confusing. Why not explicitly write out all the percentages, e.g.

From Africa

xx% in total (xx% Accumulation, xx% Aitken, xx% marine)

Alternatively explain, why you combined the Aitken and marine types. However, I find stating all percentages is the best way. Also can't be table 3 and 4 combined?

**Page 32:** Figure 8 shows the wind speed vs $N_{total}$, showing no or weak correlation. How does the wind direction correlate with $N_{total}$.

**Page 33:** The Wind speed panel in Figure 9 shows a quite different trend between 2010 and 2011. How can this be explained.

**References:**

Kirkby, J., et al. "Role of sulphuric acid, ammonia and galactic cosmic rays in atmospheric aerosol nucleation." Nature 476.7361 (2011): 429-433.

Merikanto, J., et al. "Impact of nucleation on global CCN." Atmospheric Chemistry and Physics 9.21 (2009): 8601-8616.
* * *

---

## Referee Comment (RC2) · Anonymous Referee #2 · 10 Jun 2016

The authors present measurements of aerosol size distributions, total and cloud condensation nuclei (CCN) number concentrations, and derived hygroscopicities conducted during November 2010 and April 2011 during the CARRIBA field campaign. Aerosol size distributions show distinct Aitken and accumulation modes, which are used to classify the aerosol into one of three types: 1) marine-type (both modes of same order), 2) Aitken-type (both modes present but with a pronounced Aitken mode), 3) accumulation-type (both modes present but with a pronounced accumulation mode). Ten-day air mass backtrajectories are used infer the origins of these aerosol types and show that the accumulation-type aerosols are transported via easterly flow across the Atlantic from Africa, while the Aitken-type aerosols follow more northerly trajectories (hailing sometimes from North America or the N. Atlantic). The origins of marine-type aerosol are indeterminate with transport via both easterly and northeasterly flows.

[Figure]

Comparisons of ground-based and airborne CCN number concentrations show excellent agreement and that assuming a constant aerosol hygroscopity of 0.66 combined with Kohler theory and the measured dry particle size distributions is generally sufficient to predict CCN number concentrations to within 0-15% uncertainty. In general, the manuscript is well written and the results should be of interest to the readers of ACP. I recommend publication after the following comments are satisfactorily addressed:

1) It is stated on Line 19 (and elsewhere in the manuscript, e.g., Pg. 17, Line 10) that sea spray does not contribute noticeably to Ntotal or NCCN. This strong conclusion seems to be based on a bulk average hygroscopicity of 0.66 that is lower than that for pure sodium chloride and more consistent with that for sulfate aerosols, as well as a lack of correlation between the aerosol number concentrations and local wind speeds measured 17 m above the tops of the Barbados cliffs. This does not seem to me to be strong evidence on which to base this conclusion.

Recently, Modini et al., examined similar-looking marine size distributions to those shown in Figure 1 that were measured in the Eastern Pacific during the EPEACE campaign. Their distributions also possessed the distinct Aitken and accumulation modes seen in this study with a long tail out to larger than 1um in diameter. They carried out a three-lognormal-mode fit to apportion the influence of primary marine aerosol versus the smaller size modes of likely secondary origin. Looking at the lower panels in Figure 1 of the present paper, a hump is clearly visible with a peak between 300-400 nm diameter. It is suggested in Appendix 3 that this hump may be an artifact caused by the SMPS-OPC splicing at 230-250 nm diameters. Alternatively, it could be the real manifestation of the primary marine aerosol mode. The authors should examine this further and, either way, employ the canonical size distribution fitting procedure of Modini et al. in order to quantify the likely contribution of primary marine aerosol, rather than just saying that it is negligible. It may also be worth bringing in the PDI size distributions to further constrain the supermicron tail of the dry aerosol distribution in Figure 1.

2) I don't understand the relevance of Figure 8 and the discussion related to wind

speeds measured at the tops of the Barbados cliffs. Presumably the surface wind speeds over the ocean and along the air mass backtrajectories are more relevant for aerosol production via whitecapping and bubble bursting. I suggest that this figure and discussion be removed.

3) What is the height of the Ragged Point aerosol inlet above sea level? How does this compare to the heights of the SCL sampled by ACTOS (reported as between 100-400 m)? If cloud base is at 500 m, do the ACTOS measurements indicate that the aerosol concentrations at 100 m are just as representative of the SCL as those at 400 m (i.e, the marine boundary layer is always well mixed)?

4) On Page 16, Lines 6-7, it is suggested that sea spray particles are found predominantly in the super-micron size range, which suggests a distinct mode being measured by the PDI rather than just the tail of the distribution; however, no data between 500-1000 nm are presented nor are the actual PDI size distributions shown. Please include a figure showing the ACTOS OPC distributions out to 2.5 um and the supermicron PDI distributions for each aerosol type or add them to Figure 1. This additional data should be very useful for carrying out the 3-mode fit requested in Point #1 above.

5) Figure 7 is quite nice for showing the origins of the air masses arriving at the measurement station, but is lacking any vertical context that would distinguish between free tropospheric transport of, e.g., SAL-influenced air versus more low-level transport that could be influenced by primary marine emissions and wet deposition. Please add some panels giving the altitude vs. time history of the trajectories for each aerosol type.

6) On Pg. 18, Lines 1-3, reference is drawn to both mineral dust and biomass burning particles. Is this suggesting that either of these aerosol types contribute to those sampled during CARRIBA? Until this brief discussion, I was under the impression that this was mainly a sulfate story with perhaps a minor contribution from primary marine aerosols. Please clarify.

Minor Comments:

a) What do the dotted lines (versus solid lines) in Figure 7 represent?

b) Pg. 2, Line 29: Do you mean "sea spray" rather than "sea salt"?

c) Pg. 7, Line 9: Strike redundant "the"

d) Pg. 7, Line 20: Remove comma between "both, heights"

e) Pg. 11, Line 9, 10: Change "extend" to "extent"

f) Pg. 12, Line 10: "discusses" to "discussed"

g) Pg. 13, Line 1: cmˆ3 to cmˆ-3

h) Pg. 14, Line 5: "weather" to "whether"

i) Pg. 14, Lines 10-12: Strike sentences "No correlation. . .from sea-spray"

j) Pg. 15, Line 9: "with the PDI were found in"

k) Pg. 16, Line 6: strike "and were found predominantly in the super-micron size range". This statement is unsupported as the PDI only measures in the supermicron size range.

l) Pg. 17, Line 10: Strike "A correlation. . .significantly to Ntotal"

m) Pg. 17, Line 32: Replace "towards" with "versus"

n) Pg. 18, Line 5: What is meant by "found temporarily and fragmented"? Please clarify/reword.

---

## Referee Comment (RC3) · Anonymous Referee #3 · 14 Jun 2016

The manuscript presents aerosol properties measured on Barbados, comparing ground-based results with airborne data during the CARRIBA field campaign (2010-2011). Measurements of the particle size distributions as well as the cloud condensation nuclei concentrations are shown and compared. Differences in the particles' properties are found depending on the air mass regions and were characterized for three typical cases found on the Caribbean island: marine type, Aitken type and accumulation type. This subdivision was based on ten-days back-trajectories calculated using the FLEXTRA model. Also the chemical composition, its influence on hygroscopicity and CCN activity were investigated. It was found that the knowledge of a mean hygroscopicity parameter and a time-resolved dry size distribution are enough to predict CCN concentrations within 15% uncertainty. The data and uncertainties regarding the specific set-ups are discussed extensively and the manuscript is well-structured. Therefore, I recommend it for publication after minor revisions.

**General comments:**

1) All supersaturation values occurring in the CCNCs lack a corresponding measurement uncertainty. As this is crucial for the further retrieved kappa- values it would be interesting to know the numbers and how much they influence the kappa-value calculations. This is also missing in Figure 2 where supersaturations are plotted.

2) It is mentioned that for the airborne data-set the size distributions were measured using an SMPS in combination with an OPC. Could you comment on which index of refraction was used for the OPC diameter retrieval? Also, in Figure 1 it should be mentioned that the ground based measurements are only from SMPS data, while the airborne results comprise a combination of the two techniques. I would suggest to explicitly delineate the SMPS and OPC regions in the plot for the airborne data-set.

3) In the discussion of the sea spray influence the authors state that primarily super-micron particles contained sea salt. It would be really helpful to see the OPC size distributions up to a diameter of 2.5 micrometers and investigate whether the super-micron particles appear in these measurements as well.

4) Hygroscopicity parameters (kappa) are discussed for different air masses and super-saturations. However, in section 3.2, where results are presented, no uncertainties are mentioned regarding the single kappa values and it is not specified for which size the values were retrieved (or if it is a mean over all sizes?). Later in the manuscript (Page 10, line 6) it is stated that no size dependent trend was found. This information should be shifted to the beginning of the chapter. It would be helpful to plot also the size dependence in Figure 2 as a subplot. With regard to the uncertainties, they would also help understand statements like "results agree within uncertainty" which are mentioned several times.

5) The authors state that ground based and airborne data are well comparable and that results found at the ground represent the aerosol in the sub cloud layer at Barbados (see Section 4). However, one has to be careful with this comparison as the data which is referred to in this case was measured during 1 pm and 3 pm UTC. A recent publication showed that the aerosol properties in the lower PBL are strongly dependent on time of day as the mixing of the layers

takes time and therefore a fully mixed PBL can only be found later in the day (see Rosati et al., 2016). Therefore, it would be advisable to mention the time period that was compared in this statement as the results in the morning at the ground might not have been representative for the overall marine sub cloud layer.

6) Figures 5, 6, 8 and 10 need legends to be better and more easily understood. Also, figures with subplots should be numbered as a), b) or equivalent.

**Specific comments:**

Page 2, line 9: please cite the actual chapter 7 of the IPCC instead of the whole work

Page 4, line 8: the acronym "SALTRACE" is introduced but not explained

Page 4, line 11: the acronym "CARRIBA" should be explained when it's first mentioned!

Page 6, line 3: replace "total CCN number" with "polydisperse CCN number" if this is actually meant. Could you also give a size range that can be measured by the mini-CCNC?

Page 7, line 3: replace "again and again" with "repeating"

Page 7, line 9: delete extra "the"

Page 9, line 18: data is plural; replace "was" with "were"

Page 12, line 4: replace "summarized" with "summarizes"

Page 13, line 1: wrong units are given for the dN/dlogDp concentrations

Page 14, line 5: replace "weather" with "whether"

Figure 2: replace "besides" with "except"

Figure 5: the error bars were omitted for the "left" panel but it would be recommended to at least give an approximate value in the figure caption

Figure 6: the background colors grey and orange are not explained in the caption.

---

## Author Comment (AC1) · 16 Sep 2016

**Answer to Review by Anonymous Referee #1**

We thank the reviewer for taking the time to read our manuscript carefully and thoughtfully, for commenting on it and for making valuable suggestions. Our answers to the comments are given in blue print, while your original comments are shown in black. Changes in the text are highlighted in the .pdf of the manuscript attached at the end of this review - additional text is shown in bold script, text which will be deleted is now displayed in cyan (however, small technical corrections were not highlighted to avoid a confusing appearance of the text).

This manuscript investigated the aerosol arriving at Barbados and found 3 typical number size distributions (NSD), classified as marine-type, Aitken-type and accumulation type. Size distributions on ground where compared with airborne measurements. Hygroscopicty and CCN number concentrations were investigated, also in regards of their origin. Sea spray was found to be a minor source of CCN. The manuscript is well written and structured. Measurement uncertainties and data quality are well documented and possible limitations and alternative interpretations are discussed. I recommend publication, after minor revision.

**General Comments:**

1. The reader would strongly benefit, if all Figures had a legend. Figures 2, 5, 6, 7, 8 do not have a legend, while others have.

Thank you for pointing this out. The figures have legends now.

2. The discussion of the particle hygroscopicity is very informative, however, I did not find a comparison of the different kappa values for the different NSD types. Did the different types show different or similar kappa-values? Did all Aitken/Nucleation mode types show a similarly high kappa, or was there a variation indicating different nucleation precursor? Also how representative is the kappa value for these Aitken mode types, as the maximum in the NSD is below the critical diameter. Can these kappa values be used to get an indication how these particles were formed, in terms of aerosol precursors.

It had been mentioned in the original manuscript (second paragraph of Section 3.2):

"The retrieved $\kappa$-values scattered much. No clear trends could be seen, neither when $\kappa$ was separated according to different air masses (not shown here; for a discussion of the different air masses see Sec. 3.3), nor when values for different supersaturations were examined separately (as shown in Figure 2)."

As these different air masses correspond to the different NSD types, this answers one of the points you raise here, but to make it clearer, we added "i.e., to the different types of NSD" to the previously cited sentence. Also, the different critical diameters derived for the different supersaturations are now given in the text, ranging roughly from 180 nm to 55 nm (page (P) 9, line (L) 32-33). But as there were not trends that we could discover, this does not add much to the interpretation of the particle composition. The only information that can be drawn from the values of $\kappa$ is, that particles in the respective size range do not consist of large fractions of sea salt nor organics, which has been discussed in the original manuscript already. You are correct that for the Aitken mode particles, the composition cannot directly be derived from $\kappa$. This is mentioned now explicitly in the text (P 15, L 30-32).

**Specific Comments**:

**Page 3 Lines 17ff.:** It is noted that nucleation in the free troposphere is derived from DMS-derived $H_2SO_4$. Kirkby et al. showed that pure binary $H_2SO_4$-$H_2O$ nucleation cannot explain ambient formation rates and that it requires ternary $NH_3$-$H_2SO_4$-$H_2O$ nucleation even in the free troposphere. This is nowhere mentioned or discussed. These findings should at least be mentioned.

You are right, and we know the work by Kirkby et al. (2011) and other work from that group very well (Heike Wex is a co-author on this paper). However, we feel it would be somewhat of a stretch to include this here, as our work does not explicitly treat new particle formation, and other factors influencing new particle formation: radiation, temperature, humidity or the newly discovered influence of organics (Kirkby et al., 2016) are also not explicitly mentioned, either. This would lead off the tracks we are following here. Also, all the other work cited here is explicitly related to marine aerosol (either by being based on measurements in respective environments or by modeling marine aerosol), which the work presented in Kirkby et al. (2011) is not. Hence we prefer to not include this paper.

**Page 5 Line 8:** How long were the filters stored and at which temperature before measurement. Could this influence the results?

Filters were stored for several weeks up to months at -4°C (the temperature is now given in the text, P 5, L 30). Storing filters frozen prior to analysis, sometimes for years, is the standard procedure for filters not only taken at Ragged Point, but also for many other filter samples taken by different groups worldwide. An influence, if properly done, cannot be totally excluded but also is not expected, particularly not for stable inorganic substances as those analyzed for our study (e.g., Na, Cl, sulfate and ash).

**Page 5 Line 19:** What is the absolute (wet) cut-off of the cyclone, were all droplets removed?

The cut-off of the cyclone (i.e., were 50% of all particles are removed) is 525 nm (see Figure 13, formerly lower panel of Figure 10). The corresponding growth factors that were determined for the particle hygroscopicity at a relative humidity (RH) between roughly 80% and 90% (as prevailing in the cyclone during usage at Ragged Point) had been given with 1.6 to 1.945 in Appendix A3. This results in the dry cut-off of approximately 275 nm, as given at the location you relate to, in the text. The information regarding the "true" and "wet" cut-off of the cyclone has now been explicitly added to the text in Appendix A3 (P 23, L 4 ff). As for the sizes above which all particles, haze particles or droplets are removed, this had been given in the text at the location you are referring to, for a dry diameter of 500 nm, corresponding to a size of the haze particles present in the cyclone of roughly 800 to 900 nm.

**Page 5 Line 29:** Which neutralizer was used, Krypton, X-ray? Which charging efficiency was applied?

It was a Krypton-neutralizer, and the charging efficiency was taken from Wiedensohler (1988). This is explicitly stated in the text now (P 6, L 17).

**Page 10 Lines 26ff.:** During Aitken-type events $N_{total}$ is often twice as high as $N_{CCN}$. What does this imply on the importance of new particle formation to CCN production?

Previous modelling studies (e.g. Merikanto et al., 2009) suggested that up to 45

Unfortunately, the final part of your comment was missing, but it is assumed that you meant that "45% of global low-level cloud CCN at 0.2% supersaturation are secondary aerosol derived from nucleation" (this quote is taken from the abstract of Merikanto et al., 2009). In the paper you refer to here, it is also stated that the remaining CCN come from primary emissions, which, in our case, can originate from continental sources (e.g., mineral dust and biomass burning aerosol from the Sahara and Sahel, respectively, and others) but, likely to a much lesser extent, also from sea spray. Based on our data set, it is difficult to determine to what extent exactly primary and secondary particles contribute to particles in the CCN size range. For this, a Lagrangian type data set would be better suited.

But we now included the paper by Merikanto et al. (2009) in the introduction (P 4, L 6-9), and additionally refer to it where we discuss the different aerosol types (P 15, L 33 ff).

**Page 11 Section 3.3.2:** The authors suggest that Aitken-mode NSD might have entrained from the free troposphere, as also previously suggested by several studies. Did you find any indication of this in FLEXTRA? Additionally, FLEXPART dispersion mod-elling (https://www.flexpart.eu/) would have been my choice instead of its predecessor FLEXTRA. Why was it not used?

FLEXTRA was used due to the data being easily available, and it is not to be expected that results from these two different models would yield fundamentally different results. The trajectories were allowed to originate from heights up to 3000 m at their starting points over the Sahara and Sahel region in Africa and had no heights restrictions elsewhere. They then descended down to 500 m, their heights upon arrival at Barbados. The observed particles were formed up to 3 days prior to detection and could, after initial formation, grow in size both in the free troposphere but also below. Therefore, it is difficult to determine the location of new particle formation based on the data presented in the here discussed study, which is why our statements on this are formulated as suggestions. Nothing changed.

**Page 27 Table 3 and Table 4.** I find these two tables somewhat confusing. Why not explicitly write out all the percentages, e.g. From Africa xx% in total (xx% Accumulation, xx% Aitken, xx% marine) Alternatively explain, why you combined the Aitken and marine types. However, I find stating all percentages is the best way. Also can't be table 3 and 4 combined?

One point we wanted to make in the text is, to show to which extent aerosol arriving on Barbados is influenced by continental emissions from Africa, and vice versa, how often elevated CCN number concentrations can be traced back to continental emissions from Africa. Hence, marine-type and Aitken-type, which can be interpreted as a sub-groups of the marine-type, were summarized. And then, adding the value which makes up the remainder to 100% just makes the table larger and maybe even more confusing. We did not want to split the information further, and instead, as you suggested, now explain in the text why the marine- and Aitken-type were summarized for the presented analysis (P 12, L 32 ff).

Summarizing the two tables was not done, as one used the trajectories as a base, while the other one is based on the number size distributions.

**Page 32:** Figure 8 shows the wind speed vs $N_{total}$, showing no or weak correlation. How does the wind direction correlate with $N_{total}$.

Average prevailing wind directions were 108° (+/- 24°) and 104° (+/- 20°) during November 2010 and April 2011, respectively, i.e., wind came stably from south-east, where the only exception was a time with low wind speeds during DOY 109 to 110. General wind directions were mentioned in the manuscript before. It can also be seen when looking at the trajectories (albeit admittedly not very well), that air masses arriving at Ragged Point mostly came from very similar directions. Hence, not surprisingly, a correlation of $N_{total}$ with wind direction also gives no correlation with air mass types. We mention the above given average wind directions and the fact that there was no correlation between wind direction and $N_{total}$ explicitly in the text now (P 15, L 12-13 and 19-21).

**Page 33:** The Wind speed panel in Figure 9 shows a quite different trend between 2010 and 2011. How can this be explained.

There is really no correlation between wind speed and $N_{total}$. A simple linear fit for November 2010 results in a coefficient of determination ($R^2$) of 0.001 and Pearson's correlation coefficient (P) of 0.28, and for April 2011 $R^2$ of 0.29 and P of < 0.0001 is obtained. Therefore, talking about a trend is also not correct. What you might be referring to is the fact that there are more data at higher values of $N_{total}$ in November 2010 as there were comparably more times with Aitken-type and accumulation-type air masses then, compared to April 2011 when the marine-type air masses with comparably low $N_{total}$ prevailed. Nevertheless, for both, November 2010 and April 2011, values for $N_{total}$ spread over a large range for any detected wind speed. Nothing was changed in the text

[revised manuscript text omitted]

---

## Author Comment (AC2) · 16 Sep 2016

We thank the reviewer for taking the time to read our manuscript carefully and thoughtfully, for commenting on it and for making valuable suggestions. Our answers to the comments are given in blue print, while your original comments are shown in black. Changes in the text are highlighted in the .pdf of the manuscript attached at the end of this review - additional text is shown in bold script, text which will be deleted is now displayed in cyan (however, small technical corrections were not highlighted to avoid a confusing appearance of the text).

The authors present measurements of aerosol size distributions, total and cloud condensation nuclei (CCN) number concentrations, and derived hygroscopicities conducted during November 2010 and April 2011 during the CARRIBA field campaign. Aerosol size distributions show distinct Aitken and accumulation modes, which are used to classify the aerosol into one of three types: 1) marine-type (both modes of same order), 2) Aitken-type (both modes present but with a pronounced Aitken mode), 3) accumulation-type (both modes present but with a pronounced accumulation mode). Ten-day air mass backtrajectories are used infer the origins of these aerosol types and show that the accumulation-type aerosols are transported via easterly flow across the Atlantic from Africa, while the Aitken-type aerosols follow more northerly trajectories (hailing sometimes from North America or the N. Atlantic). The origins of marine type aerosol are indeterminate with transport via both easterly and northeasterly flows.

Comparisons of ground-based and airborne CCN number concentrations show excellent agreement and that assuming a constant aerosol hygroscopity of 0.66 combined with Kohler theory and the measured dry particle size distributions is generally sufficient to predict CCN number concentrations to within 0-15% uncertainty. In general, the manuscript is well written and the results should be of interest to the readers of ACP.

I recommend publication after the following comments are satisfactorily addressed:

1) It is stated on Line 19 (and elsewhere in the manuscript, e.g., Pg. 17, Line 10) that sea spray does not contribute noticeably to Ntotal or NCCN. This strong conclusion seems to be based on a bulk average hygroscopicity of 0.66 that is lower than that for pure sodium chloride and more consistent with that for sulfate aerosols, as well as a lack of correlation between the aerosol number concentrations and local wind speeds measured 17 m above the tops of the Barbados cliffs. This does not seem to me to be strong evidence on which to base this conclusion.

Recently, Modini et al., examined similar-looking marine size distributions to those shown in Figure 1 that were measured in the Eastern Pacific during the EPEACE campaign. Their distributions also possessed the distinct Aitken and accumulation modes seen in this study with a long tail out to larger than 1um in diameter. They carried out a three-lognormal-mode fit to apportion the influence of primary marine aerosol versus the smaller size modes of likely secondary origin. Looking at the lower panels in Figure 1 of the present paper, a hump is clearly visible with a peak between 300-400 nm diameter. It is suggested in Appendix 3 that this hump may be an artifact caused by the SMPS-OPC splicing at 230-250 nm diameters. Alternatively, it could be the real manifestation of the primary marine aerosol mode. The authors should examine this further and, either way, employ the canonical size distribution fitting procedure of Modini et al. in order to quantify the likely contribution of primary marine aerosol, rather than just saying that it is negligible. It may also be worth bringing in the PDI size distributions to further constrain the supermicron tail of the dry aerosol distribution in Figure 1.

We followed your suggestion, examining the different modes of the number size distributions (NSDs) separately, for which we used the NSDs exemplarily shown in Figure 1. The "hump" (referring to your wording) referred to in the Appendix is not the same as that seen in Figure 1 (please compare the two - the former is seen when dividing number size distributions measured at Ragged Point by those simultaneously measured on ACTOS, while the latter, as you suggest, is a third particle mode which was measured on ACTOS, but not at Ragged Point due to using a cyclone). We never meant to disregard the well visible third

particle mode that formerly was, admittedly, only barely visible in Figure 1, but admittedly, we had ignored it. Now we extended the presented size range in this figure up to 2.5 μm, so this mode is now much better visible. The total masses determined when integrating over the combined SMPS-OPC NSDs agree with those determined from the filter measurements to within 73% to 114%, giving confidence in the masses derived from the NSDs.

The third mode indeed can be assumed to correspond to the primary marine aerosol (PMA) mode as described in Modini et al. (2015), at least for air masses of the Aitken- and marine-type. Particles in this mode make up roughly 90% of the total particulate mass, while they only account for 4% (Aitken-type) to 7% (marine-type) and 10% (accumulation-type) of the total particle number. Modini et al. (2015) find that PMA particles contribute 24-28% of CCN at supersaturations below 0.3% during conditions of comparably high wind speeds of (on average) 16 m/s, while they only contribute at most 5-10% for a time with average wind speeds of 12 m/s. Wind speeds during our measurements almost always were below 10 m/s, so our results are well in agreement with results from Modini et al. (2015). We included these additional results and the comparison with Modini et al. (2015) in our text and also revised occurrences of remarks concerning the contribution of sea spray particles to total and CCN number concentrations accordingly (see particularly page(P) 9, line(L) 18-22, where the analysis is introduced). Additionally, the discussion of sea spray aerosol was extended in the introduction (P 3, L 2 ff).

2) I don't understand the relevance of Figure 8 and the discussion related to wind speeds measured at the tops of the Barbados cliffs. Presumably the surface wind speeds over the ocean and along the air mass backtrajectories are more relevant for aerosol production via whitecapping and bubble bursting. I suggest that this figure and discussion be removed.

Wind speed is the relevant parameter determining the production of sea spray aerosol. We observed wind speed in the marine boundary layer (or Sub Cloud Layer, SCL) in distances up to 10 km upwind of Barbados (Siebert et al., 2013) and that on Ragged Point to be the same (see figure inserted below, which is also included in the manuscript now). One point we make in our study is, that conditions at the tower at Ragged Point are similar to those for the SCL upwind of Barbados, due to the aerosol at Ragged Point being samples on top of an elevated (30 m high) cliff on top of a mast standing out 17 m above that already elevated ground. It is not to be expected that wind speeds upwind of our measurement area (including ACTOS flights) were different for some range to come. Due to this and the agreement between the wind speeds measured in the SCL upwind of Barbados and on Ragged Point, it can also be assumed, that the wind speeds we measured were also relevant for the sea spray aerosol arriving at Barbados, which is indicated by the correlation between wind speed and mass of Na$^+$+Cl$^-$ presented in the original manuscript (now Fig 11).  Therefore, the figure you refer to adds information which is corroborating the point that while sea spray aerosol may add a considerable fraction of the total aerosol mass, it only makes up a much smaller small fraction of the total number concentration (see also our answer to your point 1) and does not correlate to number concentrations observed in the first two aerosol modes. Nothing changed.

[Figure]

3) What is the height of the Ragged Point aerosol inlet above sea level? How does this compare to the heights of the SCL sampled by ACTOS (reported as between 100-400 m)? If cloud base is at 500 m, do the ACTOS measurements indicate that the aerosol concentrations at 100 m are just as representative of the SCL as those at 400 m (i.e, the marine boundary layer is always well mixed)?

The short answer to your question is: "Yes!"

The heights of the aerosol inlet (as given in the manuscript before) it is 17 m above the ground, where the latter is on top of a 30 m high cliff (the heights of the cliff is given in the text now, too, P 5, L 27). Aerosol properties measured on ACTOS at different heights in the SCL (e.g., during the profiles) were the same throughout the whole SCL, and also compare well with what was measured on Ragged Point. That was discussed extensively in the original manuscript, for parameters as total and CCN number concentrations and size distributions. It indeed seems that the marine boundary layer was always well mixed, down to the height of the aerosol inlet at Ragged Point. This is not too surprising, as e.g., a diurnal variation (as observed for the continental boundary layer) is not to be expected, due to the comparably stable sea surface temperature (near to no cooling occurs during the night due to the large body of water). We refer to this now in the text (L 7, L 10-14).

Other than that, we think that it was extensively mentioned that the good agreement between measurements on ACTOS and at Ragged Point justify the assumption that the marine boundary layer was always well mixed during our measurements (e.g., Abstract, L 3-4; P 8, L 20-22; P 11, L 13 ff (to end of section) etc.). Hence nothing else was changed.

4) On Page 16, Lines 6-7, it is suggested that sea spray particles are found predominantly in the super-micron size range, which suggests a distinct mode being measured by the PDI rather than just the tail of the distribution; however, no data between 500-1000 nm are presented nor are the actual PDI size distributions shown. Please include a figure showing the ACTOS OPC distributions out to 2.5 um and the supermicron PDI distributions for each aerosol type or add them to Figure 1. This additional data should be very useful for carrying out the 3-mode fit requested in Point #1 above.

The number size distributions on ACTOS measured with the optical particle counter (OPC) are now shown up to a diameter of 2.5 $\mu$m in Figure 1 and also the three mode fit was done (see our answer to your point 1 above).

5) Figure 7 is quite nice for showing the origins of the air masses arriving at the measurement station, but is lacking any vertical context that would distinguish between free tropospheric transport of, e.g., SAL-influenced air versus more low-level transport that could be influenced by primary marine emissions and wet deposition. Please add some panels giving the altitude vs. time history of the trajectories for each aerosol type.

A respective figure and description is now included (P 14, L 12 ff).

6) On Pg. 18, Lines 1-3, reference is drawn to both mineral dust and biomass burning particles. Is this suggesting that either of these aerosol types contribute to those sampled during CARRIBA? Until this brief discussion, I was under the impression that this was mainly a sulfate story with perhaps a minor contribution from primary marine aerosols. Please clarify.

We are somewhat puzzled about this remark. The original manuscript already discussed the contributions from the different aerosols in some length. This started in the abstract (and is still there, P 1, L 7-9), continued in the introduction with introducing the Saharan Air Layer (SAL, now on P 4, L 18 ff) and had a thorough discussion in Sec. 3.3.2 "Origin of the different aerosol types" and Sec. 3.3.3 "Discussion of the different aeosol types". We exemplarily want to refer you to the first two paragraphs of the latter, which did start and are still starting with:

"Aerosol of the accumulation-type often occurred during times when increased amounts of insoluble material were analyzed on the daily filter samples taken at Ragged Point (see top panel of Figure 7, denoted as ash). The corresponding increase in $N_{total}$ and $N_{CCN}$ can hence be expected to originate from

biomass burning or desert dust particles, i.e., from particles of continental origin. This is in agreement with the above discussed origin of the respective air masses in the Sahara or Sahel region in Africa. …".

and ending with:

"... Such large particle number concentrations in this size range were not observed at other times. This is indicative for mineral dust particles from the Sahara with respective sizes being present during this time. "

As we feel that this topic was treated thoroughly throughout the original manuscript, nothing was changed.

Minor Comments:

a) What do the dotted lines (versus solid lines) in Figure 7 represent?

The "lines" are actually single dots, one for every calculation (i.e., one every 4 hours), and when the dots are separately visible, this means that the air masses traveled further within 4 hours then when they fall together. This is mentioned in the figure caption now.

b) Pg. 2, Line 29: Do you mean "sea spray" rather than "sea salt"?

We changed the sentence in question to: "Aerosol particles from a marine source, often called sea spray or sea salt particles, are generally produced in dependence on surface wind speeds over oceans."

c) Pg. 7, Line 9: Strike redundant "the" Done

d) Pg. 7, Line 20: Remove comma between "both, heights" Done

e) Pg. 11, Line 9, 10: Change "extend" to "extent" Done

f) Pg. 12, Line 10: "discusses" to "discussed" Done

g) Pg. 13, Line 1: cm^3 to cm^-3 Done

h) Pg. 14, Line 5: "weather" to "whether" Done

i) Pg. 14, Lines 10-12: Strike sentences "No correlation. . .from sea-spray"

As explained above (mainly our response to your point 2 and also to your point 1), the statements you want us to delete are valid and we will not change them. We did, however, change arguments concerning the contributions of sea salt particle throughout the text where appropriate. Following the statement you refer to, here, we now say:" This is in agreement with the earlier reported fraction of only 4 to 10% that particles in the observed sea spray mode contribute to $N_{total}$ (see Sec. 3.1)." (see P 16, L 26-27).

j) Pg. 15, Line 9: "with the PDI were found in" Done

k) Pg. 16, Line 6: strike "and were found predominantly in the super-micron size range". This statement is unsupported as the PDI only measures in the supermicron size range.

The paragraph containing this sentence was changed significantly (striking this statement), please check at P 18, L28ff.

l) Pg. 17, Line 10: Strike "A correlation. . .significantly to Ntotal"

Modified to: "A correlation of $N_{total}$ to wind speed was, however, not observed, which is in accordance to the fact that particles originating from sea spray were found to only contribute a few percent to $N_{total}$ in accordance also with results by Modini et al. (2015)." (see P 20, L 8-10)

m) Pg. 17, Line 32: Replace "towards" with "versus" We replaced "towards" with "compared to".

n) Pg. 18, Line 5: What is meant by "found temporarily and fragmented"? Please clarify/reword.

We replaced "fragmented" by "spatially patchy", taking this expression from the paper we cite at the respective paragraph (Hamilton et al., 2014).

Literature:

[revised manuscript text omitted]

---

## Author Comment (AC3) · 16 Sep 2016

**Answer to Review by Anonymous Referee #3**

We thank the reviewer for taking the time to read our manuscript carefully and thoughtfully, for commenting on it and for making valuable suggestions. Our answers to the comments are given in blue print, while your original comments are shown in black. Changes in the text are highlighted in the .pdf of the manuscript attached at the end of this review - additional text is shown in bold script, text which will be deleted is now displayed in cyan (however, small technical corrections were not highlighted to avoid a confusing appearance of the text).

The manuscript presents aerosol properties measured on Barbados, comparing ground-based results with airborne data during the CARRIBA field campaign (2010-2011). Measurements of the particle size distributions as well as the cloud condensation nuclei concentrations are shown and compared. Differences in the particles' properties are found depending on the air mass regions and were characterized for three typical cases found on the Caribbean island: marine type, Aitken type and accumulation type. This subdivision was based on ten-days back-trajectories calculated using the FLEXTRA model. Also the chemical composition, its influence on hygroscopicity and CCN activity were investigated. It was found that the knowledge of a mean hygroscopicity parameter and a time-resolved dry size distribution are enough to predict CCN concentrations within 15% uncertainty. The data and uncertainties regarding the specific set-ups are discussed extensively and the manuscript is wellstructured. Therefore, I recommend it for publication after minor revisions.

**General comments:**

1) All supersaturation values occurring in the CCNCs lack a corresponding measurement uncertainty. As this is crucial for the further retrieved kappa- values it would be interesting to know the numbers and how much they influence the kappa-value calculations. This is also missing in Figure 2 where supersaturations are plotted.

Uncertainty in supersaturations were based on numerous (> 30) separate calibrations which had been done at our home laboratory (TROPOS), in accordance with the ACTRIS protocol (http://www.actris.net/Portals/97/Publications/quality%20standards/aerosol%20insitu/WP3_D3.13_M24_CCNC_SOP_v130514.pdf). Based on this, supersaturations together with one standard deviation are: 0.07 (+/- 0.0033) %, 0.1 (+/- 0.0033) %, 0.2 (+/- 0.0067) %, 0.3 (+/- 0.01) %, 0.4 (+/- 0.0133) %, i.e., 3.3% (relative) for supersaturations of or above 0.1% and 3.3% (absolute) for a supersaturation of 0.07%. If these uncertainties were shown in Figure 2, they would mostly vanish within the symbols, so we only added them to the text (see page(P) 10, line(L) 8-9). The uncertainty in the derived kappa-values originating from the uncertainty in the supersaturation is roughly 1/3 of that obtained from averaging the data, a fact which is mentioned in the text now, too (P 10, L 9).

2) It is mentioned that for the airborne data-set the size distributions were measured using an SMPS in combination with an OPC. Could you comment on which index of refraction was used for the OPC diameter retrieval? Also, in Figure 1 it should be mentioned that the ground based measurements are only from SMPS data, while the airborne results comprise a combination of the two techniques. I would suggest to explicitly delineate the SMPS and OPC regions in the plot for the airborne data-set.

The refractive index is now given in Section 2 ("Measurements") where the OPC is first introduced (P 7, L 21-23). In the caption of Fig. 1 it is now explicitly stated that the airborne data-set was composed

from data of a SMPS up to 250 nm and an OPC above. We did, however, not optically highlight this difference in the figure as we feel the plot would become too busy.

3) In the discussion of the sea spray influence the authors state that primarily super-micron particles contained sea salt. It would be really helpful to see the OPC size distributions up to a diameter of 2.5 micrometers and investigate whether the super-micron particles appear in these measurements as well.

Figure 1 now shows the particle size distributions up to 2.5 micrometers, displaying the shoulder seen by the OPC clearly. Also, in answer to your comment here and to Reviewer2, estimates of the fraction of particle number and mass that was present in this mode related to the shoulder were estimated. Results are now given in the text (P 9, L 18-22).

Also, additional literature dealing with sea spray aerosol has now been included in the Introduction (P 3, L 2 ff).

4) Hygroscopicity parameters (kappa) are discussed for different air masses and supersaturations. However, in section 3.2, where results are presented, no uncertainties are mentioned regarding the single kappa values and it is not specified for which size the values were retrieved (or if it is a mean over all sizes?). Later in the manuscript (Page 10, line 6) it is stated that no size dependent trend was found. This information should be shifted to the beginning of the chapter. It would be helpful to plot also the size dependence in Figure 2 as a subplot. With regard to the uncertainties, they would also help understand statements like "results agree within uncertainty" which are mentioned several times.

The maximum uncertainty in retrieved kappa values originating from uncertainties in the supersaturations are now mentioned in the text (P 10, L 7-9, see also our answer to your comment 1). The critical diameters were, indeed, not given, as they correspond to the kappa values and are somewhat redundant. We did, however, add values for the average determined critical diameters (ranging roughly from 55 to 180 nm) to the text, now (P 9, L 31-33). We also make it clearer now that one kappa value was derived per measured number size distribution (P 9, L 31).

It had been stated in the original manuscript, that no dependence of kappa on air mass or on supersaturation was found (beginning of second paragraph of Sec. 3.2). This implies that also no dependence of the particle diameter on air mass or supersaturation can be found. But to make this clear, we now state this explicitly (P 10, L 1). An additional plot showing the size dependence will basically reproduce what is seen for kappa, therefore this will not be added.

The discussion concerning "results agree within uncertainty" was reworded in the paragraph in question (P 10, L 6-7).

5) The authors state that ground based and airborne data are well comparable and that results found at the ground represent the aerosol in the sub cloud layer at Barbados (see Section 4). However, one has to be careful with this comparison as the data which is referred to in this case was measured during 1 pm and 3 pm UTC. A recent publication showed that the aerosol properties in the lower PBL are strongly dependent on time of day as the mixing of the layers takes time and therefore a fully mixed PBL can only be found later in the day (see Rosati et al., 2016). Therefore, it would be advisable to mention the time period that was compared in this statement as the results in the morning at the ground might not have been representative for the overall marine sub cloud layer.

The measurement time from 1 pm to 3 pm (UTC) was given for those three days for which measured size distributions were shown, exemplarily. It is correct that most flights started roughly between 1 pm up to 3 pm (UTC), but roughly a quarter of all flights started later, as late as 7:30 pm (UTC). These times are all given in UTC, and local time on Barbados is 4 hours earlier, so flights started between 9 am up to 3:30 pm local time. This is now explained more explicitly in the instrumental section (P 7, L 8 ff). Agreement between ground based and airborne measurements was obtained independent of the time at which the flight started.

Concerning the paper you refer to: we assume you mean Rosati et al. (2016) in ACP, "Studying the vertical aerosol extinction coefficient by comparing in situ airborne data and elastic backscatter lidar"? Therefore, we have to say that it is known that the marine boundary layer is fundamentally different from a continental boundary layer (the latter is examined in Rosati et al., 2016), due to the fact that the sea surface temperature does not show such a diurnal variation as the continental surface temperatures do, and this knowledge is not new, neither for the continental nor for the marine boundary layer.

The aerosol in the marine boundary layer is rather influenced by long distance transport (as we argue in the manuscript) and (for the sea spray aerosol) by wind speed (upwind of the measurement point). The figure below, which is now also included in the manuscript, shows wind speeds measured at Ragged Point with a 30 min resolution, together with data derived from airborne measurements in the sub cloud layer (taken from Siebert et al., 2013). It illustrates that there is good agreement between wind speed measured on Ragged Point and in the sub cloud layer and that there is no diurnal cycle in wind speed.

The good agreement between airborne and ground based measured wind speed and also, as shown in the manuscript, aerosol properties (aerosol size distributions and CCN number concentrations) indicates that indeed aerosol and meteorological measurements at Ragged Point are free of influences from the island, with Ragged Point being a spot where air hits the island first after its passage over the Atlantic, and having the aerosol inlet high above the ground (on top of a 17 m high mast on top of a 30 m high cliff).

And, as said above, wind speed did not show a diurnal trend, and neither did the aerosol properties observed at Ragged Point. Also, a diurnal trend was not seen in the lidar-data obtained at Deebles Point (right next to Ragged Point), which were shown in the CARRIBA-overview-paper by Siebert et al., (2013).

Therefore, there is no reason to believe that data we took at Ragged Point are not representative of the marine sub cloud layer or that there would be a clear diurnal variation. We did, however, stress in the text that all comparisons were done at comparably similar times (within 8 hours) during the day (again, see P 7, L 8ff).

[Figure]

6) Figures 5, 6, 8 and 10 need legends to be better and more easily understood. Also, figures with subplots should be numbered as a), b) or equivalent.

Legends were added to these and more figures, and figures with 3 panels were numbered as you suggested, adjusting the text respectively.

**Specific comments:**

Page 2, line 9: please cite the actual chapter 7 of the IPCC instead of the whole work Done

Page 4, line 8: the acronym "SALTRACE" is introduced but not explained Done

Page 4, line 11: the acronym "CARRIBA" should be explained when it's first mentioned! Done (It was explained in the abstract before, now it is again given in the introduction.)

Page 6, line 3: replace "total CCN number" with "polydisperse CCN number" if this is actually meant. Done

Could you also give a size range that can be measured by the mini-CCNC?

The lowest detected size depends on the adjusted supersaturation and the hygroscopicity of the particles, so it is not clear what it exactly is that you want.

Page 7, line 3: replace "again and again" with "repeating"

We replaced it by "time and time again".

Page 7, line 9: delete extra "the" Done

Page 9, line 18: data is plural; replace "was" with "were" Done

Page 12, line 4: replace "summarized" with "summarizes" Done

Page 13, line 1: wrong units are given for the dN/dlogDp concentrations Done

Page 14, line 5: replace "weather" with "whether" Done

Figure 2: replace "besides" with "except" The caption has been modified.

Figure 5: the error bars were omitted for the "left" panel but it would be recommended to at least give an approximate value in the figure caption

Panel A of Fig. 5, the one you are referring to, would become very busy when error bars were added, so instead we now explicitly mention the typical measurement uncertainty of 20% in the figure caption.

Figure 6: the background colors grey and orange are not explained in the caption. Done

Literature:

Rosati et al. (2016), Studying the vertical aerosol extinction coefficient by comparing in situ airborne data and elastic backscatter lidar, Atmos. Chem. Phys., 16, 4539–4554, doi:10.5194/acp-16-4539-2016.

[revised manuscript text omitted]

---

## Author Response (AR2)

Dear Editor!

In the here uploaded file the technical issues you mentioned were addressed. We thank you for your efforts and input.

Regards from

Heike Wex